# GeoMamba: Geometry-Guided Recontextualization for Precise Long-Context Memory

## Abstract

Precise long-context memory, the ability to recall specific entities, bindings, and keys from long prompts, is crucial yet fragile in modern language models. Linear-time state-space models (SSMs) such as Mamba achieve efficient long-context processing via fixed-capacity hidden states, but this compression creates a memory bottleneck that degrades precise recall as sequences grow. We study a strict inference setting: frozen SSM backbones, a hard generation-time budget $B < T$ (applied only to the final inference pass), and order-preserving recontextualization. Our key insight is that selecting high-evidence tokens in causal order and generating from a shorter prompt can yield higher recall than streaming the full context through a recurrent backbone, provided that cross-window similarity is geometrically calibrated. We propose **GeoMamba**, a training-free geometry-guided pipeline that treats overlapping windows as local charts, measures pairwise transports via Orthogonal Procrustes, and scores tokens by path-covariant similarity, an invariant that remains well-defined under per-window coordinate changes. On Mamba2-1.3B, GeoMamba improves NiAH FULL8 average from 13.6 (full context) to 26.0 with $B$=2048, and LongBench-E average from 8.26 to 19.24 with $B$=4096.

## 1. Introduction

Recent advances in efficient attention and recurrence (Dao et al., 2022; Dai et al., 2019) make longer contexts practical, yet *precise long-context memory* remains fragile when evidence lies far from the query. Models perform best when relevant information appears at the beginning or end of the input, and degrade when critical evidence lies in the

[1]Anonymous Institution, Anonymous City, Anonymous Region, Anonymous Country. Correspondence to: Anonymous Author <anon.email@domain.com>.

Preliminary work. Under review by the International Conference on Machine Learning (ICML). Do not distribute.

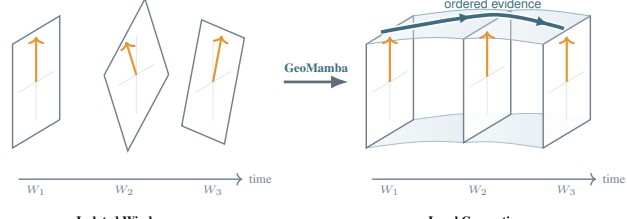

*Figure 1.* **GeoMamba transforms isolated windows into a transport-aligned manifold for ordered evidence selection.** *Left:* Isolated window encodings have misaligned local coordinates (tilted planes); early evidence fades along the sequence. *Right:* GeoMamba calibrates cross-window metrics via Procrustes transport, aligns representations, and selects an ordered evidence path (blue curve). *Takeaway:* Geometric alignment is essential; without it, cross-window similarity is unreliable.

middle (Liu et al., 2024). Linear-time State-Space Models (SSMs) such as Mamba make long contexts computationally feasible via fixed-capacity hidden states (Gu & Dao, 2024), but this compression creates a fundamental memory bottleneck: as sequences grow longer, early evidence competes with later tokens and precise recall becomes unreliable (Merrill et al., 2024). This memory–fidelity tension is largely unresolved for frozen recurrent models under hard input budgets.

This paper focuses on a stricter setting: *frozen* recurrent SSM backbones, a *hard* input budget $B < T$, and *order preservation* (no token reordering). Long-context inference becomes a *budgeted evidence selection* problem: choose an order-preserving subsequence $\tilde{x} \preceq x_{1:T}$ that retains query-relevant evidence.

Existing methods leave this setting unaddressed. Long-context extensions for SSMs, such as receptive-field enlargement (Ye et al., 2025) and length extrapolation (Ben-Kish et al., 2024), target full-context inference at longer lengths rather than budgeted prompts for a frozen backbone. Meanwhile, prompt compressors are developed for attention-based LLMs (Jiang et al., 2023; 2024; Pan et al., 2024; Xu et al., 2023; Ge et al., 2023), and retrieval-augmented methods rely on external corpora (Lewis et al., 2020); neither fits frozen recurrent SSMs where token order determines the state trajectory.

Our working hypothesis is that under a strict budget, a small set of high-evidence tokens in causal order can dominate precise recall despite the recurrent memory bottleneck. A naïve approach that scores tokens by embedding similarity fails because window-independent encoding places each window's latents in incompatible local coordinates; comparing embeddings across windows can conflate coordinate drift (the same token may "look different" across charts) with semantic difference. We resolve this by treating each window as a local chart, measuring pairwise transports via Orthogonal Procrustes on overlaps, and scoring tokens by *path-covariant similarity*. This coordinate-free notion, analogous to parallel transport on manifolds, is provably invariant to per-window basis changes.

We propose **GeoMamba**, a training-free pipeline that: (i) encodes overlapping windows independently (fresh state), (ii) canonicalizes per-window geometry via regularized whitening, (iii) measures overlap transports via weighted Procrustes, (iv) scores positions with path-covariant similarity, and (v) compresses the prompt by an order-preserving policy before re-running the frozen backbone. Section 4 proves that this score is well-defined (independent of chart choices) and yields a principled ranking of evidence positions. The main contributions are:

- **Gauge-invariant evidence scoring.** We introduce a path-covariant similarity measure that is provably invariant to per-window basis choices, enabling principled cross-window evidence ranking without global coordinate alignment.

- **Geometry-guided recontextualization.** We build overlapping window covers, canonicalize per-window representations via regularized whitening, and measure overlap transports via weighted Orthogonal Procrustes, yielding a training-free pipeline that selects an order-preserving evidence subsequence for frozen recurrent backbones.

## 2. Related Work

GeoMamba sits at the intersection of long-context SSMs, prompt compression, and representation alignment. We focus on work most relevant to our constraints: *frozen recurrent* backbones, a *hard input budget* $B < T$, and *order-preserving* recontextualization.

**Long-context state-space models.** State-space models such as S4 (Gu et al., 2021), H3 (Fu et al., 2022), and Hyena (Poli et al., 2023) achieve linear-time sequence modeling. Mamba (Gu & Dao, 2024) introduces selective state spaces, and the SSD duality (Dao & Gu, 2024) connects SSMs and attention. Related recurrent architectures include xLSTM (Beck et al., 2024), RWKV (Peng et al., 2023), RetNet (Sun et al., 2023), and hybrid SSM-attention models (Lieber et al., 2024; Ren et al., 2024). Despite efficient

inference, SSMs compress history into fixed-capacity hidden states, creating a memory bottleneck that limits *precise recall* (Merrill et al., 2024; Arora et al., 2023) (cf. "Lost in the Middle" (Liu et al., 2024) and long-context benchmarks (Bai et al., 2024; Hsieh et al., 2024; An et al., 2024; Zhang et al., 2024)). SSM extensions such as receptive-field enlargement (Ye et al., 2025) and length extrapolation (Ben-Kish et al., 2024) improve full-context accessibility to longer sequences, but do not address our strict setting of producing a budgeted prompt for a frozen recurrent backbone. GeoMamba instead targets *precise memory* under a hard budget by selecting an order-preserving subsequence and re-running the frozen backbone for generation. Unlike context-extension methods that often still operate on the full prompt, this setting requires reliable evidence selection from window-local representations, where cross-window similarity must be geometrically calibrated.

**Prompt compression and context distillation.** Prompt compression reduces input length by selecting tokens or learning compact representations. Perplexity-based selection (LLMLingua (Jiang et al., 2023; 2024), LLMLingua-2 (Pan et al., 2024)), learned compressors (RECOMP (Xu et al., 2023), In-Context Autoencoder (Ge et al., 2023)), and KV-cache eviction (Li et al., 2024; Zhang et al., 2023) are primarily developed for attention-based models. Linear attention (Katharopoulos et al., 2020) and kernel approximations (Choromanski et al., 2020) reduce attention cost, but are orthogonal to SSM-specific memory bottlenecks. More broadly, most compression methods are designed around attention backbones; GeoMamba instead outputs an *order-preserving subsequence* compatible with frozen recurrent SSMs, and scores *within-prompt* evidence via geometrically calibrated similarity on SSM hidden states.

**Retrieval-augmented generation.** RAG methods (Lewis et al., 2020) retrieve external documents to augment LLM context. Recent variants such as Self-RAG (Asai et al., 2023) improve retrieval quality through self-reflection. Unlike RAG, GeoMamba does not retrieve from an external corpus; it selects evidence *within* the given prompt under a hard budget while preserving causal order for recurrent state evolution.

**Representation alignment.** The Orthogonal Procrustes problem (Schönemann, 1966) aligns representations across views and is widely used in manifold alignment (Wang & Mahadevan, 2008) and cross-lingual embedding alignment (Grave et al., 2019). Sheaf neural networks (Bodnar et al., 2022; Hansen & Gebhart, 2020) equip nodes with local coordinates connected via transport maps (Singer & Wu, 2012). GeoMamba adapts this to *temporal* window covers: whitening (Kessy et al., 2018) calibrates per-window geometry, and Procrustes yields orthogonal transports on overlaps. We do *not* solve for a global frame; path-covariant

similarity propagates queries along connections to score evidence.

## 3. Preliminaries and Notation

We study *precise long-context memory* for frozen recurrent SSMs: recalling entities, bindings, and key–value pairs from long prompts.

### 3.1. Precise Memory and Order-Preserving Recontextualization

Let $x_{1:T} = (x_1, \ldots, x_T)$ be a token sequence of length $T$. Given a frozen language model $f$ and a task metric $\text{Eval}(\cdot, y^*)$, we consider tasks requiring *precise recall*: retrieving a "needle" sentence, tracing variable assignments, or matching key–value pairs (Weston et al., 2014; Sukhbaatar et al., 2015).

Given a hard budget $B < T$, we consider recontextualization by selecting an order-preserving subsequence $\tilde{x}$ with $|\tilde{x}| \le B$:

$$\tilde{x}^* = \arg \max_{\substack{|\tilde{x}| \le B \\ \tilde{x} \preceq x_{1:T}}} \text{Eval}(f(\tilde{x}), y^*), \tag{1}$$

where $\tilde{x} \preceq x_{1:T}$ denotes that $\tilde{x}$ is an order-preserving subsequence of $x_{1:T}$ (i.e., $\tilde{x} = (x_{t_1}, \ldots, x_{t_m})$ with $1 \le t_1 < \cdots < t_m \le T$), $y^*$ is the ground-truth response, and Eval is the task metric (e.g., exact match, F1). For recurrent backbones, token reordering changes the state trajectory, so we restrict to order-preserving selection in Eq. (1). GeoMamba may compute evidence scores via an additional preprocessing pass over the full prompt, but generation is performed on the budgeted subsequence with $|\tilde{x}| \le B$.

### 3.2. Selective SSMs and the Memory Bottleneck

SSMs such as Mamba (Gu & Dao, 2024) and Mamba-2 (Dao & Gu, 2024) process sequences via a recurrent hidden state $h_t \in \mathbb{R}^{d_{\text{model}}}$:

$$h_t = \bar{A}_t h_{t-1} + \bar{B}_t e_t, \qquad o_t = C_t h_t, \tag{2}$$

where $e_t$ is the input embedding of token $x_t$, and $\bar{A}_t, \bar{B}_t, C_t$ are input-dependent (selective) discretized matrices. The key property is *linear-time* inference: the hidden state compresses history into a fixed-capacity vector, enabling $O(T)$ complexity (Gu et al., 2021). However, this compression creates a *memory bottleneck*: early information must compete with later tokens for representation in $h_t$, and precise recall degrades as $T$ grows (Merrill et al., 2024; Arora et al., 2023).

**Implication.** This motivates inference-time control: we select an order-preserving subset of evidence tokens and

re-run the frozen backbone on the resulting prompt. In Section 4, we define a *covariant evidence score* that is well-defined under local window charts and supports principled evidence selection without any global coordinate system.

**Notation.** We use 1-based indexing ($t \in \{1, \ldots, T\}$) and treat token representations as row vectors. Let $x_{1:T}$ be the prompt and $\{W_i\}_{i=1}^n$ be overlapping windows of size $w$ with stride $s < w$. Within each window $i$, we work in a $k$-dimensional canonical chart $Z_i \in \mathbb{R}^{w \times k}$ and denote overlap transports by $U_{i \leftarrow j} \in \text{O}(k)$. An order-preserving subsequence is denoted $\tilde{x} \preceq x_{1:T}$. All remaining symbols are defined where first used; Appendix B.5 provides a consolidated reference table of core notation.

## 4. Covariant Evidence Score

GeoMamba does *not* solve for a single global window frame. Instead, it defines an evidence score using only local charts and measured overlap transports (Section 5.3).

**Idea.** Treat each window as a local chart. Using orthogonal transports on overlaps, we transport a tail content query along a maximum-reliability path and score tokens by path-covariant similarity. Intuitively, we "move" the tail query into each window's local coordinates and compare cosine similarity within that chart. Proposition 4.1 shows the resulting score is well-defined under per-window orthogonal "gauge" changes.

### 4.1. Covariant Retrieval (LAR)

**Local Alignment Retrieval (LAR)** maps canonical window latents and measured overlap transports on the cover graph to per-position scores $\text{Score}(t)$ and a chunk-expanded keep mask $m(t)$ for ordered compression (Section 5.4). It combines path-covariant similarity with lightweight, task-agnostic saliency factors.

**Tail content query.** Let $a$ be the tail (anchor) window index, and let $q_a \in \mathbb{R}^{1 \times k}$ be the mean of the last $q$ canonical token latents in $a$ (a tail-pooled content query). Here $\text{Tail}(a)$ denotes the indices of the last $q$ *valid* tokens in window $a$, and $v_a(t) \in \{0, 1\}$ is the per-token validity mask (padding and burn-in removed):

$$q_a = \frac{\sum_{t \in \text{Tail}(a)} v_a(t) \, Z_a(t)}{\sum_{t \in \text{Tail}(a)} v_a(t)}. \tag{3}$$

**Query pooling (optional).** The simplest query in Eq. (3) uses mean pooling. Unless otherwise noted, we use prompt-local IDF-weighted pooling (Eq. (7)) and a multi-query variant: we form additional anchor queries from the top-IDF tail tokens and combine their semantic scores by either a probabilistic union or max.

**Best-path query transport.** We transport $q_a$ to each window by composing transports along a single most reliable path on the cover graph. Each edge $(i, j)$ has overlap weight $\omega_{ij} \in (0, 1]$ (Eq. (16)) and an orthogonal transport $U_{i \leftarrow j} \in \mathrm{O}(k)$ measured from overlap features (Section 5.3). We define an edge cost $\ell_{ij} := -\log \omega_{ij}$ and run Dijkstra from $a$ to obtain, for every window $i$, a single shortest (maximum-reliability) path $\pi(a \to i)$. We use a single best path (rather than path averaging) to keep computation minimal; Appendix A (Lemma A.12) relates discrepancies between two path transports to loop holonomy as a diagnostic upper bound. Let $U_{i \leftarrow a}^{\pi}$ be the path-composed transport along $\pi(a \to i)$:

$$U_{i \leftarrow a}^{\pi} = U_{i_1 \leftarrow i_0} U_{i_2 \leftarrow i_1} \cdots U_{i_\ell \leftarrow i_{\ell-1}}, \quad (4)$$

where $\pi(a \to i) = (i_0 = a, i_1, \ldots, i_\ell = i)$ is the window-index sequence on the shortest path (row-vector convention). We define the transported query in window $i$ as $q_i := q_a U_{i \leftarrow a}^{\pi}$. Virtual edges, when present, are treated as additional weighted edges in the same graph.

**Semantic similarity aggregation.** For any position $t$ covered by window $i$, we compute cosine similarity in the local chart: $\mathrm{sim}_i(t) = \cos(q_i, Z_i(t))$. Let $\mathcal{C}(t)$ be the set of windows covering absolute position $t$. We aggregate by a uniform overlap average (each covering window contributes equally):

$$\mathrm{sim}(t) = \frac{1}{|\mathcal{C}(t)|} \sum_{i \in \mathcal{C}(t)} \mathrm{sim}_i(t). \quad (5)$$

We then apply a monotone activation (sigmoid) to obtain a nonnegative semantic score:

$$\mathrm{Sem}(t) = \sigma(\beta \, \mathrm{sim}(t)), \quad (6)$$

where $\beta > 0$ controls sharpness. We use a sigmoid in all experiments.

**Proposition 4.1** (Gauge invariance of the covariant evidence score). *Let $\{R_i\}_{i=1}^{n} \subset \mathrm{O}(k)$ be arbitrary per-window orthogonal basis changes, and define transformed latents $Z_i'(t) = Z_i(t) R_i$. Define transformed transports by $U_{i \leftarrow j}' = R_j^{\top} U_{i \leftarrow j} R_i$ and transformed queries by $q_a' = q_a R_a$. Then best-path transport yields $q_i' = q_i R_i$, so the resulting semantic scores $\mathrm{Sem}(t)$ are unchanged. Moreover, with the same hyperparameters $(\alpha_{\mathrm{lex}}, \gamma, \lambda_{\mathrm{heat}})$, the full evidence score $\mathrm{Score}(t)$ in Eq. (10) is unchanged for all positions $t$.*

**Proof.** See Appendix A ("Gauge invariance of the covariant evidence score").

**Auxiliary gains (optional).** The semantic score $\mathrm{Sem}(t)$ from path-covariant similarity is the core signal. Optionally, we add a lightweight lexical overlap term $\mathrm{Lex}(t)$ for discrete,

rare tokens (e.g., names, numbers) that may not be well-represented in latent space. Let $\mathcal{Q}$ be the set of unique token IDs in the query tail and $c(u)$ the count of token $u$ in the prompt; we ignore frequent tokens via a cap $f_{\mathrm{max}} = 128$ and use a prompt-local IDF proxy $\mathrm{idf}(u) = \log \frac{T+1}{c(u)+1}$:

$$g(t) = \sum_{u \in \mathcal{Q}} \mathbf{1}[0 < c(u) \leq f_{\mathrm{max}}] \, \mathrm{idf}(u) \, \mathbf{1}[x_t = u],$$
$$\mathrm{Lex}(t) = \frac{g(t)}{\max_{t'} g(t')} \in [0, 1], \quad (7)$$

We set $\mathrm{Lex}(t) = 0$ when $\max_{t'} g(t') = 0$. The base score is

$$\mathrm{Base}(t) = \mathrm{Sem}(t) + \alpha_{\mathrm{lex}} \, \mathrm{Lex}(t). \quad (8)$$

Optionally, we multiply by (i) a loss heatmap gain $\exp(\lambda_{\mathrm{heat}} L(t))$ and (ii) an innovation gain $(1 + \xi_{\mathrm{log}}(t))^{\gamma}$. In all main results we set $\lambda_{\mathrm{heat}} = 0$; when enabled, $L(t)$ and its bounds are defined in Appendix A.6. We treat $\alpha_{\mathrm{lex}}$ and $\gamma$ as task-agnostic scalars and fix them per benchmark (no per-task tuning).

**Innovation residual.** For an overlap token at absolute position $t$ covered by adjacent windows $i-1$ and $i$, let $z_{i-1}^{\mathrm{raw}}(t), z_i^{\mathrm{raw}}(t) \in \mathbb{R}^{1 \times k}$ denote its *pre-whitening* stable-dimension latents in the respective windows (rows of $Z_{i-1}^{\mathrm{raw}}$ and $Z_i^{\mathrm{raw}}$; Section 5.3). Let $\mu_i$ and $\mathbf{W}_i$ be the per-window mean and whitening map used to construct $Z_i$. We predict $z_i^{\mathrm{raw}}(t)$ from $z_{i-1}^{\mathrm{raw}}(t)$ via the affine transport:

$$M_{i-1 \to i} = \mathbf{W}_{i-1} U_{i \leftarrow i-1} \mathbf{W}_i^{-1},$$
$$\hat{z}_i(t) = (z_{i-1}^{\mathrm{raw}}(t) - \mu_{i-1}) M_{i-1 \to i} + \mu_i. \quad (9)$$

The residual $\xi(t) = z_i^{\mathrm{raw}}(t) - \hat{z}_i(t)$ measures cross-window disagreement after accounting for the overlap transport. We log-compress its norm as $\xi_{\mathrm{log}}(t) = \log(1 + \|\xi(t) \mathbf{W}_i\|_2)$, average across overlaps when applicable, and set $\xi_{\mathrm{log}}(t) = 0$ outside overlaps. Intuitively, large $\xi_{\mathrm{log}}(t)$ indicates residual cross-window disagreement after alignment.

The final per-position score is

$$\mathrm{Score}(t) = \mathrm{Base}(t) \cdot \exp(\lambda_{\mathrm{heat}} L(t)) \cdot (1 + \xi_{\mathrm{log}}(t))^{\gamma}. \quad (10)$$

**Chunk-expanded keep mask.** We convert pointwise saliency into contiguous evidence chunks. In *token* mode, we select top-$K$ seed positions by $\mathrm{Score}(t)$ (excluding the query tail) and expand each seed by a fixed radius $r$:

$$S = \mathrm{Top}\text{-}K\{\,\mathrm{Score}(t) \mid t \notin \mathrm{tail}\,\},$$
$$m(t) = \mathbf{1}\big[\exists u \in S : |t - u| \leq r\big]. \quad (11)$$

In *span* mode, we instead select fixed-length spans by an energy detector,

$$E(u) = \sum_{t=u}^{u+\ell-1} \mathrm{Score}(t), \quad (12)$$

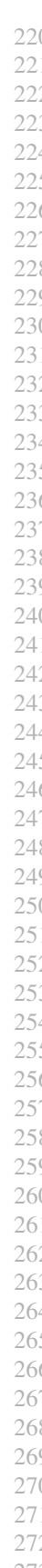

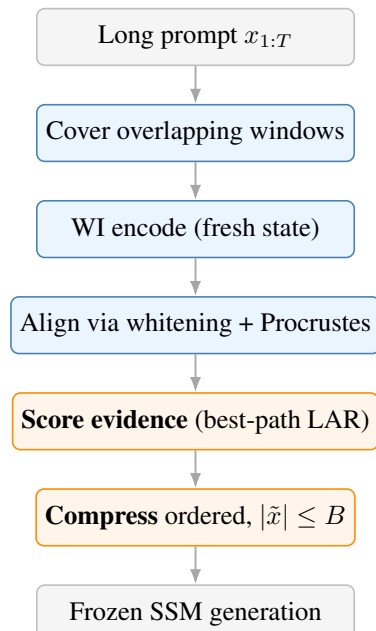

Long prompt $x_{1:T}$

↓

Cover overlapping windows

↓

WI encode (fresh state)

↓

Align via whitening + Procrustes

↓

**Score evidence** (best-path LAR)

↓

**Compress** ordered, $|\tilde{x}| \leq B$

↓

Frozen SSM generation

*Figure 2.* **GeoMamba pipeline.** Given a long prompt $x_{1:T}$ and budget $B$, we construct an order-preserving compressed prompt $\tilde{x}$ and run the frozen backbone. *Method:* Cover overlapping windows, encode independently (fresh state), canonicalize with stable-dimension whitening, estimate overlap transports by Procrustes, score tokens via best-path LAR, and compress in order under $B$. *Takeaway:* The geometry module (align + score) is the core innovation; it calibrates cross-window coordinates before evidence selection.

and keep the union of the top-$K$ non-overlapping spans. Optionally, we keep the smallest subset of top spans whose cumulative energy exceeds a target fraction (a simple posterior competition). Here $m(t)$ denotes the keep mask, distinct from the per-window validity masks $v_i(t)$ used to remove padding and burn-in tokens.

# 5. GeoMamba

## 5.1. Pipeline Overview

GeoMamba is a training-free recontextualization pipeline: it (i) builds a local geometric view of a long prompt from overlapping windows, (ii) scores evidence tokens with a covariant retrieval signal (Section 4.1), and (iii) compresses the prompt under strict order preservation before running the frozen backbone for generation.

Figure 2 gives a compact overview of the primary pipeline, and Algorithm 1 provides pseudocode. GeoMamba is training-free and only changes the *input prompt* via ordered compression.

**Complexity (single prompt).** Let $n$ be the number of windows. WI encoding costs $n$ backbone runs on length-

**Algorithm 1** GeoMamba — training-free recontextualization (primary pipeline)

---
**Require:** Prompt tokens $x_{1:T}$, budget $B$, window size $w$, stride $s$
**Ensure:** Compressed prompt $\tilde{x}$ (order-preserving subsequence of $x_{1:T}$)
1: $\{W_i\}_{i=1}^n \leftarrow$ Partition$(x_{1:T}, w, s)$; build cover graph $G = (V, E, \omega)$ {Cover graph construction}
2: $H_i^{\text{bb}} \leftarrow$ SSM$(W_i)$ independently (fresh state) {Window-independent encoding}
3: $Z_i \leftarrow$ StableDims+Whiten$(H_i^{\text{bb}})$ {Shared stable subspace + whitening}
4: $U_{i \leftarrow j} \leftarrow$ WeightedProcrustes$(Z_i[\Omega_{ij}], Z_j[\Omega_{ij}], \Gamma_{ij})$ for $(i, j) \in E$ {Overlap transports (weighted Procrustes)}
5: $(\text{Score}(t), m(t)) \leftarrow$ LAR$(Z, U, G)$ {Covariant retrieval scoring (LAR)}
6: $\tilde{x} \leftarrow$ OrderedCompress$(x_{1:T}, \text{Score}(t), m(t), B)$ {Ordered compression (no reordering)}
7: **return** $\tilde{x}$

---

$O(w)$ sequences; transports require $O(|E| \, k^3)$ SVDs plus overlap multiplications; best-path transport is one Dijkstra run ($O(|E| \log n)$). Ordered compression is linear in $T$ plus chunk sorting. GeoMamba is inference-time only and trades extra preprocessing compute ($O(nw)$ tokens, often $> T$ due to overlap) for improved recall under strict budgets.

**Reproducibility pointer.** Implementation details and default hyperparameters are provided in Appendix B.8.

## 5.2. Problem setup and hard constraints

We study long-context inference for a frozen causal backbone (SSM/Mamba): given a prompt $x_{1:T}$ and a strict input budget $B < T$, we construct a budgeted prompt $\tilde{x}$ with $|\tilde{x}| \leq B$ to maximize the task metric under greedy decoding.

**Hard constraint 1: causal monotonicity.** We restrict $\tilde{x}$ to be an *ordered subsequence* of the original prompt:

$$\tilde{x} = (x_{t_1}, x_{t_2}, \ldots, x_{t_{|\tilde{x}|}}),$$
$$1 \leq t_1 < \cdots < t_{|\tilde{x}|} \leq T. \tag{13}$$

This ensures the backbone is exposed to retained evidence in the same temporal order as in the original prompt. Any token reordering changes the input sequence and thus alters the recurrent state trajectory; we focus on ordered-only recontextualization and do not consider non-ordered injection or reordering.

**Hard constraint 2: no global coordinates.** We avoid global synchronization: we estimate only pairwise overlap transports and compose them along paths, keeping the score well-defined under local charts without solving for a single global frame.

## 5.3. Geometry construction: local charts and measured connections

We now detail the geometry construction. Raw latents $Z_i^{\text{raw}} \in \mathbb{R}^{w \times k}$ are obtained by selecting $k$ stable original

dimensions from $H_i^{\text{bb}}$; per-window whitening yields canonical latents $Z_i$ (details below). For overlap edge $(i, j)$, the transport $U_{i \leftarrow j} \in \text{O}(k)$ satisfies $Z_i[\Omega_{ij}] \approx Z_j[\Omega_{ij}] \, U_{i \leftarrow j}$.

**Coverage graph.** We consider a length-$T$ token sequence $x_{1:T}$, partitioned into $n$ overlapping windows $\{W_i\}_{i=1}^n$ of fixed size $w$ with stride $s < w$: Overlaps provide redundant views of the same tokens, which we exploit to estimate transports on overlaps and to average token scores across multiple coverings.

$$W_i := \{a_i, \ldots, \min(a_i + w - 1, T)\},$$

$$a_i := 1 + (i-1)s, \qquad n := \max\left(1, \ 1 + \left\lceil \frac{T-w}{s} \right\rceil\right). \tag{14}$$

When $T < w$ we have a single window $W_1 = \{1, \ldots, T\}$. In the implementation we pad boundary windows to length $w$ and mask padding and burn-in tokens in all geometry statistics; we reserve $W_i$ for token-index sets and boldface $\mathbf{W}_i$ for whitening maps. For window pairs, we denote the overlap index set

$$\Omega_{ij} := W_i \cap W_j. \tag{15}$$

The *coverage graph* $G = (V, E, \omega)$ has vertices $V = \{1, \ldots, n\}$ (window indices) and edges $E$ between pairs with sufficient overlap. Edge weights encode overlap strength:

$$\omega_{ij} = \frac{|W_i \cap W_j|}{\min(|W_i|, |W_j|)}. \tag{16}$$

We connect windows when $\omega_{ij}$ exceeds an overlap threshold (we use $\tau = 0.1$ in all experiments). We treat $\omega_{ij}$ as an overlap reliability signal: higher overlap yields more stable transport estimation and lower path cost in best-path query transport.

**Measured connections.** In a sheaf/bundle view, each window is a local chart and overlap agreement is mediated by transports; here the connection $U_{i \leftarrow j}$ is *measured* from overlap features (Hansen & Gebhart, 2020; Bodnar et al., 2022). This keeps the pipeline training-free and auditable: $U_{i \leftarrow j}$ is computed analytically from overlap features rather than learned by an additional alignment network.

**Window-independent encoding.** Each window is encoded *independently* by the frozen SSM backbone, starting from a fresh recurrent state. This reduces long-range state drift: local representations depend primarily on local context, making overlap agreement measurable and stable. To reduce boundary transients, we optionally prepend a short warm-start prefix of length $P_{\text{ws}}$ from the immediate left context (main runs use $P_{\text{ws}}{=}0$; 0 disables warm-start):

$$H_i^{\text{bb}} = f\big(x_{a_i - P_{\text{ws}} : a_i + w - 1}\big)\big|_{a_i : a_i + w - 1} \in \mathbb{R}^{w \times d_{\text{model}}}, \tag{17}$$

where $f$ is the frozen backbone and we keep only hidden states aligned to the window tokens. We also apply a burn-in mask that excludes the first $b$ tokens of each window from geometry statistics (mean/covariance, stable-dimension selection, and transport estimation).

**Shared stable dimensions.** To define a shared transport space without per-window basis drift, we select $k$ *original* backbone dimensions shared across all windows by masked variance. Using original dimensions (instead of learned per-window projections) keeps the coordinate choice explicit and makes transports comparable across windows. Let $v_i(t) \in \{0, 1\}$ be the per-token validity mask within window $i$ (padding and burn-in removed). For each dimension $j \in \{1, \ldots, d_{\text{model}}\}$, define a per-window variance

$$\text{Var}_i(j) = \text{Var}_t\big(H_i^{\text{bb}}(t, j) \mid v_i(t) = 1\big),$$

aggregate conservatively $\text{Var}(j) = \min_i \text{Var}_i(j)$, favoring dimensions that remain active and stable across all windows, and select indices

$$\mathcal{D} = \text{Top-}k \, \text{Var}(j), \qquad Z_i^{\text{raw}} = H_i^{\text{bb}}[:, \mathcal{D}] \in \mathbb{R}^{w \times k}. \tag{18}$$

**Metric calibration (whitening).** We canonicalize each window by subtracting a masked mean and applying per-window regularized ZCA whitening. This reduces per-window scale and correlation differences, improving conditioning for overlap alignment and making cosine similarities comparable across windows. Let

$$\mu_i = E[Z_i^{\text{raw}}], \qquad \Sigma_i = E\big[(Z_i^{\text{raw}} - \mu_i)^\top (Z_i^{\text{raw}} - \mu_i)\big],$$

where expectations are over valid tokens in window $i$. Let $\Sigma_i = Q_i \, \text{diag}(\lambda_i) \, Q_i^\top$ be an eigendecomposition and define $\lambda_{\text{reg}, i} := \rho \cdot \max\{\max_k \lambda_{i,k}, \epsilon\}$ with a small $\epsilon > 0$ (per-window; $\rho$ fixed) to avoid degenerate covariances. We define

$$\begin{aligned} \mathbf{W}_i &= Q_i \, \text{diag}\big((\lambda_i + \lambda_{\text{reg}, i})^{-1/2}\big) \, Q_i^\top, \\ Z_i &= (Z_i^{\text{raw}} - \mu_i) \, \mathbf{W}_i, \end{aligned} \tag{19}$$

so that canonical features have comparable scale across windows. We also keep $\mathbf{W}_i^{-1}$ for affine-connection residuals used by the scoring function.

**Analytic transport via (weighted) Procrustes.** We need to compare token embeddings across windows using cosine similarity, but WI encoding places each window's latents in a different coordinate system. Orthogonal Procrustes is the unique transformation that (i) minimizes Frobenius distance on overlap features and (ii) preserves inner products, which is exactly the invariant signal we need. We thus solve a weighted Orthogonal Procrustes problem on $\Omega_{ij}$:

$$U_{i \leftarrow j} = \arg\min_{U \in \text{O}(k)} \big\| \Gamma_{ij}^{1/2} \big( Z_i[\Omega_{ij}] - Z_j[\Omega_{ij}] \, U \big) \big\|_F, \tag{20}$$

where $\Gamma_{ij} = \mathrm{diag}(v_i(t) \cdot v_j(t))_{t \in \Omega_{ij}}$ weights each overlap token by the product of its validity masks in both windows (excluding padding and burn-in). Let the weighted cross-covariance be

$$\mathbf{S}_{ij} := Z_j[\Omega_{ij}]^\top \, \Gamma_{ij} \, Z_i[\Omega_{ij}] \in \mathbb{R}^{k \times k}.$$

If $\mathbf{S}_{ij} = P \Sigma Q^\top$ is an SVD, then one minimizer is

$$U_{i \leftarrow j} = P Q^\top \in \mathrm{O}(k). \tag{21}$$

In our implementation, transports are stored as $U_{i \leftarrow j}$ (mapping chart $j \to i$).

**Virtual edges (optional).**  We optionally add non-adjacent *virtual edges* between semantically similar windows, providing shortcut paths that reduce error accumulation when composing many transports. Virtual edges are selected by window-level cosine similarity in canonical space; details and hyperparameters are in Appendix B.8.

**Covariant evidence score (LAR).**  Given canonical latents and measured transports, we compute the covariant evidence score $\mathrm{Score}(t)$ and keep mask $m(t)$ as defined in Section 4.1.

### 5.4. Ordered Path Compression

In ordered compression, given the keep mask and $\mathrm{Score}(t)$ from covariant retrieval (Section 4.1), we construct an order-preserving compressed prompt $\tilde{x}$ under the budget constraint $|\tilde{x}| \leq B$. This step turns pointwise evidence scores into a concrete prompt: we keep the instruction/question context and then allocate the remaining budget to high-scoring evidence in causal order.

**Order preservation (no token reordering).**  GeoMamba never injects tokens out of order. Instead, we keep a subset of tokens from the original prompt and output them in the original time order.

**Chunk selection with scaffolds.**  We always keep a short tail (and optionally a prefix), and fill the remaining budget by selecting high-scoring contiguous chunks induced by the keep mask. Chunk-based selection reduces fragmentation compared to selecting isolated tokens, and scaffolds can preserve minimal continuity when the mid-region is heavily compressed. Let $P_{\mathrm{keep}}$ and $T_{\mathrm{keep}}$ be the prefix/tail lengths, and let $B_{\mathrm{mid}} := B - P_{\mathrm{keep}} - T_{\mathrm{keep}}$ be the remaining budget for the mid region. We convert the mid-region keep mask into contiguous chunks $\mathcal{K} = \{[b_k, e_k]\}$ and rank each chunk by its sum score:

$$p_k = \sum_{t \in [b_k, e_k)} \mathrm{Score}(t).$$

We then greedily select chunks in descending $p_k$ order under the budget constraint $\sum_{k \in \mathcal{S}} (e_k - b_k) \leq B_{\mathrm{mid}}$. To maintain continuity, we optionally add sparse scaffold anchors at a fixed stride inside the mid region. If budget remains, we fill the remaining mid budget with a single contiguous span centered at the highest-scoring remaining position, avoiding fragmented single-token additions (then emit all kept indices in increasing order). Finally, we output $\tilde{x}$ by ordered concatenation of all kept indices (no reordering).

**Proposition 5.1** (Feasibility of ordered compression)**.** *The ordered compression policy outputs a prompt $\tilde{x}$ that is an order-preserving subsequence of the original prompt ($\tilde{x} \preceq x_{1:T}$) and satisfies the hard budget constraint $|\tilde{x}| \leq B$. Moreover, if $B \geq T$, ordered compression leaves the sequence unchanged: $\tilde{x} = x_{1:T}$.*

**Proof.** See Appendix A ("Ordered compression: feasibility under a hard budget").

## 6. Experiments

We evaluate GeoMamba as an *inference-time* recontextualization method for frozen SSM backbones. Our goal is to study *precise long-context memory* under hard *input-budget* constraints without any additional training. We focus on two complementary benchmarks: RULER NiAH FULL8 at context length 4096 (synthetic key/value retrieval) and LongBench-E (e-13) at context length 16384 (real-world multi-task long-context understanding). Tables 1 and 2 report the main results for GeoMamba on a frozen Mamba2-1.3B backbone, comparing against full-context Mamba2 and LongMamba baselines. Appendix C provides additional results on Mamba2-130M (including geometry ablations and budget sensitivity) and an external DeciMamba reference.

### 6.1. Experimental Settings

**Datasets.**  We evaluate on RULER NiAH FULL8 (ctx=4096; 8 tasks) and LongBench-E (e-13; ctx=16384; 13 English tasks), reporting exact-match accuracy (%) and scores in $[0, 100]$, respectively.

**Backbone, baselines, and protocol.**  We evaluate GeoMamba on a frozen Mamba2-1.3B backbone. Baselines include the full-context Mamba2-1.3B model and LongMamba-1.3B (full context) as a long-context SSM baseline. GeoMamba generates from an order-preserving compressed prompt with budgets $B{=}2048$ (NiAH) and $B{=}4096$ (LongBench-E) using benchmark-provided prompts/scorers and greedy decoding.

**Cost accounting.** The budget $B$ refers to the number of tokens fed to the frozen backbone in the *final generation pass*. GeoMamba's preprocessing performs window-independent encoding over the original prompt (and may process more

*Table 1.* **RULER NiAH (FULL8, ctx=4096) on Mamba2-1.3B.** Task accuracy (%). Trunc = head+tail truncation under the same budget $B$=2048. *Takeaway:* GeoMamba significantly outperforms both full-context baselines and same-budget truncation.

| Model | Method | S1 | S2 | S3 | MK1 | MK2 | MK3 | MV | MQ | Avg. |
|---|---|---|---|---|---|---|---|---|---|---|
| Mamba2-1.3B | Vanilla | 100.0 | 1.6 | 2.8 | 2.8 | 0.6 | 0.2 | 0.5 | 0.1 | 13.6 |
| Mamba2-1.3B | LongMamba | 100.0 | 42.4 | 21.4 | 16.0 | 0.0 | 0.2 | 6.8 | 8.8 | 24.4 |
| Mamba2-1.3B | Trunc ($B$) | 50.6 | 35.2 | 36.6 | 21.2 | 0.8 | 0.6 | 9.5 | 12.25 | 20.84 |
| Mamba2-1.3B | **GeoMamba** | 99.0 | **47.4** | **26.0** | **18.6** | **1.6** | **0.0** | 5.0 | 10.8 | **26.0** |

*Table 2.* **LongBench-E (e-13, ctx=16384) on Mamba2-1.3B.** Scores on the 13-task English subset. Trunc = head+tail truncation under the same budget $B$=4096. *Takeaway:* GeoMamba outperforms full-context baselines and truncation (+35% relative).

| Model | Method | Synthetic | | Summary | | Single-doc QA | | Multi-doc QA | | Few-shot Learning | | | Coding | | Avg. |
|---|---|---|---|---|---|---|---|---|---|---|---|---|---|---|---|
| | | PC | PR | GR | MN | MQA | QA | 2WM | HQA | SS | TR | TQA | LCC | RB | |
| Mamba2-1.3B | Vanilla | 1.18 | 0.81 | 7.61 | 11.34 | 5.66 | 2.24 | 2.28 | 2.79 | 4.64 | 14.33 | 9.88 | 23.13 | 21.45 | 8.26 |
| Mamba2-1.3B | LongMamba | 1.48 | 5.32 | 14.87 | 14.72 | 10.98 | 5.00 | 5.37 | 5.34 | 14.02 | 22.33 | 48.04 | 41.83 | 34.53 | 17.22 |
| Mamba2-1.3B | Trunc ($B$) | 0.89 | 0.97 | 10.02 | 12.62 | 6.96 | 2.67 | 3.96 | 3.89 | 7.46 | 33.67 | 17.50 | 48.12 | 36.60 | 14.26 |
| Mamba2-1.3B | **GeoMamba** | 0.93 | 4.87 | 15.15 | 17.08 | 13.71 | 4.82 | 8.79 | 6.27 | 11.27 | 39.00 | 39.78 | 49.56 | 38.94 | 19.24 |

than $T$ tokens due to overlap), so we do not claim compute parity with full-context baselines; "25–50% tokens" refers to generation input length rather than total compute.

**Hyperparameter selection.** We select a single configuration per benchmark on a fixed calibration split (held out from the reported evaluation split) and freeze it for all reported runs. For NiAH (FULL8), we tune on $n$=200 samples per task; for LongBench-E, we tune on $n$=200 examples from a representative 4-task subset (Qasper, HotpotQA, PassageRetrieval-en, RepoBench-P) to keep calibration compute manageable. All hyperparameters are reported in Table 4; selection protocol is in Appendix C.1. **Default pipeline configuration.** Unless otherwise noted, main experiments use IDF-weighted query pooling with a multi-query variant (top-4 IDF tokens, union combination), span-based seed selection (span length 128), and loss heatmap *disabled* ($\lambda_{\text{heat}}$=0). These choices are frozen before evaluation and shared across all tasks within each benchmark.

### 6.2. Main Results: RULER Precise-Memory Tasks (NiAH)

We evaluate *precise long-context memory* on the eight NiAH variants in RULER, which range from single-needle retrieval (S1–S3) to multi-key/value binding (MK1–MK3, MV, MQ). Table 1 reports per-task accuracy for Mamba2-1.3B and LongMamba (full context) and GeoMamba. Overall, GeoMamba improves NiAH FULL8 average from 13.6 (full context) to 26.0 with $B$=2048 (50% of ctx), significantly outperforming same-budget head+tail truncation (20.84).

We report per-task scores because NiAH variants can fail in qualitatively different ways: multi-key binding tasks (MK*) stress *discrete* key/value consistency, while single-needle tasks (S*) are closer to targeted retrieval under distractors.

The largest gain is on S2 (1.6 → 47.4); S1 is near ceiling and MK2/MK3 remain near zero at this scale. This aligns with our hypothesis: single-needle tasks benefit most under a hard budget, while multi-key binding remains difficult for single-pass order-preserving selection.

### 6.3. Main Results: LongBench-E (e-13)

We evaluate LongBench-E on its 13-task English subset at 16K context. Table 2 reports task scores grouped by LongBench-E categories. GeoMamba improves over the full-context baseline (8.26 → 19.24) with $B$=4096 (25% of ctx), outperforming both LongMamba (17.22) and same-budget truncation (14.26, +35% relative). For example, TREC (TR) improves from 14.33 to 39.00, and long code completion (LCC) improves from 23.13 to 49.56. The largest gains come from code and classification tasks (LCC/RB/TR), consistent with sparse-evidence settings, while gaps versus LongMamba remain on TriviaQA and summarization (TQA/SS) and passage count (PC); passage retrieval (PR) improves substantially. Ablations on Mamba2-130M (Appendix C.2) confirm that geometry is not a single-trick effect: removing transports, whitening, or best-path query transport each degrades the average by about 3–4%.

## 7. Conclusion

We study precise long-context memory for state-space models from a geometric calibration perspective. We formulate recontextualization as order-preserving subsequence selection under a hard budget, and present GeoMamba, a training-free pipeline that canonicalizes window-local charts via whitening and calibrates cross-window similarity via Procrustes transport. Experiments on RULER and LongBench-E show significant gains over baselines and naïve truncation.

## Impact Statement

This work improves precise long-context memory for state-space models, enabling applications such as document understanding, code analysis, and question answering over long inputs. However, enhanced retrieval capabilities may amplify dual-use risks: surveillance systems could more effectively extract personal information from large corpora, and misinformation tools could selectively retrieve misleading evidence. We recommend restricting high-stakes deployment to audited settings, implementing content filtering for sensitive domains, and transparently reporting failure modes. The token-level selection scores produced by Geo-Mamba may facilitate human oversight of which evidence is retained under hard budgets.

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

# Appendix

**Appendix organization.** The appendix is structured in three sections: **A. Proofs**, **B. Background and notation** (including task-name abbreviations and implementation notes), and **C. Additional experiments**.

## A. Proofs

**Scope (what this appendix proves).** Following common practice (stating key propositions in the main text and deferring full proofs), this appendix provides self-contained proofs for the core theoretical claims used in the main body: (i) closed-form (weighted) Procrustes measurement and its gauge-equivariance (Lemmas A.3–A.5); (ii) gauge invariance of the covariant evidence score $\mathrm{Score}(t)$ under per-window orthogonal basis changes; (iii) a conservative holonomy bound quantifying discrepancy between two path transports.

**Notation convention (rows vs. columns).** In the main paper we stack token vectors as rows, so $H_i[\Omega] \in \mathbb{R}^{|\Omega| \times k}$. For linear-algebraic Procrustes proofs we use the equivalent column-stacked form $A_i := H_i[\Omega]^\top \in \mathbb{R}^{k \times |\Omega|}$. Translating back to the paper convention simply inserts transposes.

**Proof template.** We keep all steps explicit by (i) restating each claim with explicit shapes, (ii) isolating reusable inequalities (e.g., von Neumann's trace inequality), and (iii) matching the same row-vector transport convention as the main paper.

### A.1. Auxiliary: window cover properties

**Lemma A.1** (Open cover of a length-$T$ sequence). *Let $T \in \mathbb{N}$, window size $w \in \mathbb{N}$, and stride $s \in \mathbb{N}$ with $w \geq s > 0$. Define window start indices $a_i := 1 + (i-1)s$ and windows*

$$W_i := \{a_i, \ldots, \min(a_i + w - 1, T)\} \subset \{1, \ldots, T\}, \qquad i = 1, \ldots, n, \tag{22}$$

*where $n := \max\left(1, \ 1 + \left\lceil \frac{T-w}{s} \right\rceil\right)$ (as in Eq. (14)). Then $\{W_i\}_{i=1}^n$ forms an open cover of $\{1, \ldots, T\}$, i.e., for every position $t \in \{1, \ldots, T\}$ there exists at least one window index $i$ such that $t \in W_i$.*

*Proof.* Fix any $t \in \{1, \ldots, T\}$. Let

$$i_0 := \left\lfloor \frac{t-1}{s} \right\rfloor + 1.$$

Set $i := \min(i_0, n)$. Then $i \in \{1, \ldots, n\}$ and

$$a_i = 1 + (i-1)s \leq 1 + (i_0 - 1)s \leq t.$$

If $i = i_0$, then $t \leq a_i + s - 1 \leq a_i + w - 1$ since $w \geq s$. If $i = n$, then $t \leq T \leq a_n + w - 1 = a_i + w - 1$ by the definition of $n$. Therefore $a_i \leq t \leq \min(a_i + w - 1, T)$, i.e., $t \in W_i$. □

**Lemma A.2** (Adjacent-overlap connectivity condition). *Assume $w > s > 0$ and let the overlap threshold satisfy*

$$\tau \leq \frac{w-s}{w}. \tag{23}$$

*Then the cover graph built by connecting any window pair whose overlap ratio exceeds $\tau$ contains all adjacent edges $(i, i+1)$ and is connected.*

*Proof.* Consider adjacent windows in the interior, where $|W_i| = |W_{i+1}| = w$. Their overlap size is $|W_i \cap W_{i+1}| = w - s$, so the overlap ratio equals $(w-s)/w$. By the threshold assumption, the ratio exceeds $\tau$, so the adjacent edge is present. Since all adjacent edges are present, the window indices form a path graph, hence the cover graph is connected. □

### A.2. Procrustes measurement (closed form)

**Lemma A.3** (Procrustes via SVD). *Let $A, B \in \mathbb{R}^{k \times m}$. Consider the Orthogonal Procrustes problem*

$$U^\star = \arg \min_{U \in \mathrm{O}(k)} \|B - UA\|_F^2. \tag{24}$$

*Let $BA^\top = P\Sigma Q^\top$ be an SVD. Then one minimizer is $U^\star = PQ^\top \in \mathrm{O}(k)$.*

*Proof.* Expanding the Frobenius norm gives

$$\|B - UA\|_F^2 = \|B\|_F^2 + \|UA\|_F^2 - 2\operatorname{tr}(U^\top BA^\top). \tag{25}$$

Since $U$ is orthogonal, $\|UA\|_F = \|A\|_F$, so minimizing the left-hand side is equivalent to maximizing $\operatorname{tr}(U^\top BA^\top)$ over $U \in \mathrm{O}(k)$. Write $BA^\top = P\Sigma Q^\top$. Then

$$\operatorname{tr}(U^\top BA^\top) = \operatorname{tr}\big((P^\top UQ)^\top \Sigma\big). \tag{26}$$

Let $\tilde{U} := P^\top UQ \in \mathrm{O}(k)$. By von Neumann's trace inequality, $\operatorname{tr}(\tilde{U}^\top \Sigma) \leq \operatorname{tr}(\Sigma)$, with equality when $\tilde{U} = I$. This is achieved by $U = PQ^\top$. $\square$

**Lemma A.4** (Weighted Procrustes via SVD). *Let $A, B \in \mathbb{R}^{k\times m}$ and let $W = \operatorname{diag}(w_1, \ldots, w_m)$ with $w_t \geq 0$. Consider the weighted Orthogonal Procrustes problem*

$$U^\star = \arg\min_{U\in\mathrm{O}(k)} \left\|(B - UA)W^{1/2}\right\|_F^2. \tag{27}$$

*Define $\tilde{A} := AW^{1/2}$ and $\tilde{B} := BW^{1/2}$. Let $\tilde{B}\tilde{A}^\top = P\Sigma Q^\top$ be an SVD. Then one minimizer is $U^\star = PQ^\top \in \mathrm{O}(k)$.*

*Proof.* By definition of $\tilde{A}, \tilde{B}$ and invariance of the Frobenius norm,

$$\|(B - UA)W^{1/2}\|_F^2 = \|\tilde{B} - U\tilde{A}\|_F^2. \tag{28}$$

This reduces the weighted problem to the standard Procrustes problem in Lemma A.3 with $(A, B)$ replaced by $(\tilde{A}, \tilde{B})$. $\square$

## A.3. Procrustes gauge equivariance

**Lemma A.5** (Procrustes is gauge-equivariant). *Let $R_i, R_j \in \mathrm{O}(k)$ be per-window gauge changes and suppose the overlap matrices transform as $H_i'[\Omega] = R_i H_i[\Omega]$ and $H_j'[\Omega] = R_j H_j[\Omega]$ (token vectors viewed as columns). Let $U_{ij}$ be any Procrustes minimizer for $(H_i[\Omega], H_j[\Omega])$ in Equation (20). Then $U_{ij}' := R_j U_{ij} R_i^\top$ is a Procrustes minimizer for $(H_i'[\Omega], H_j'[\Omega])$.*

*Proof.* Let $W \succeq 0$ be any diagonal weight over overlap tokens (in the main paper, $W$ corresponds to $\Gamma_{ij}$). Write the (weighted) Procrustes objective as

$$f(U) = \left\|\big(H_j[\Omega] - UH_i[\Omega]\big)W^{1/2}\right\|_F^2 \quad \text{over } U \in \mathrm{O}(k).$$

Under the gauge change, the transformed objective is

$$f'(U) = \left\|\big(R_j H_j[\Omega] - U\, R_i H_i[\Omega]\big)W^{1/2}\right\|_F^2. \tag{29}$$

Using orthogonality and Frobenius invariance under left multiplication, we have

$$f'(U) = \left\|\big(H_j[\Omega] - R_j^\top U R_i\, H_i[\Omega]\big)W^{1/2}\right\|_F^2 = f(\tilde{U}), \tag{30}$$

where $\tilde{U} := R_j^\top U R_i \in \mathrm{O}(k)$. Thus minimizing $f'$ over $U$ is equivalent to minimizing $f$ over $\tilde{U}$. If $U_{ij}$ minimizes $f$, then choosing $U = R_j U_{ij} R_i^\top$ yields $\tilde{U} = U_{ij}$ and therefore minimizes $f'$. $\square$

## A.4. Best-path transport (maximum reliability)

**Lemma A.6** (Shortest path on $-\log$ weights equals maximum product). *Let $G = (V, E)$ be a graph with positive edge weights $\omega_e \in (0, 1]$. Define additive edge costs $\ell_e := -\log\omega_e$ and for any path $\pi$ define*

$$C(\pi) := \sum_{e\in\pi} \ell_e, \qquad R(\pi) := \prod_{e\in\pi} \omega_e. \tag{31}$$

*Then for any fixed endpoints $u, v \in V$,*

$$\arg\min_{\pi:u\to v} C(\pi) = \arg\max_{\pi:u\to v} R(\pi). \tag{32}$$

*Proof.* For any path $\pi$, we have

$$C(\pi) = \sum_{e \in \pi} -\log \omega_e = -\log\left(\prod_{e \in \pi} \omega_e\right) = -\log R(\pi). \tag{33}$$

Since $-\log(\cdot)$ is strictly decreasing on $(0, \infty)$, minimizing $C(\pi)$ is equivalent to maximizing $R(\pi)$. $\square$

### A.5. Proof of Proposition 4.1 (gauge invariance)

This subsection provides full proofs for Proposition 4.1 in the main text.

**Setup.** We follow the row-vector convention of the main paper. Let $Z_i(t) \in \mathbb{R}^{1 \times k}$ denote canonical token latents in window $i$, and let $U_{i \leftarrow j} \in \mathrm{O}(k)$ be measured transports (mapping chart $j \to i$ by right multiplication). Fix an anchor window $a$ with query $q_a \in \mathbb{R}^{1 \times k}$. For a fixed path $\pi(a \to i)$, define the path-composed transport $U_{i \leftarrow a}^\pi$ and transported query $q_i$:

$$U_{i \leftarrow a}^\pi = \prod_{(u \leftarrow v) \in \pi(a \to i)} U_{u \leftarrow v}, \qquad q_i := q_a\, U_{i \leftarrow a}^\pi. \tag{34}$$

**Lemma A.7** (Equivariance of best-path query transport). *Let $\{R_i\}_{i=1}^n \subset \mathrm{O}(k)$ be per-window orthogonal basis changes. Define transformed latents $Z_i'(t) = Z_i(t)\, R_i$ and transformed transports $U_{i \leftarrow j}' = R_j^\top U_{i \leftarrow j} R_i$. Let the transformed anchor query be $q_a' = q_a R_a$ and define transported queries by the same path rule:*

$$q_i' := q_a'\, U_{i \leftarrow a}'^{\,\pi}. \tag{35}$$

*Then $q_i' = q_i R_i$ for every window $i$.*

*Proof.* It suffices to prove an edge-wise recursion. Consider an edge $(u \leftarrow v)$ and assume $q_v' = q_v R_v$. Then

$$q_u' = q_v' U_{u \leftarrow v}' = (q_v R_v)(R_v^\top U_{u \leftarrow v} R_u) = q_v U_{u \leftarrow v} R_u = q_u R_u. \tag{36}$$

By induction along the chosen path, we obtain $q_i' = q_i R_i$ for all $i$. $\square$

**Lemma A.8** (Gauge invariance of path-covariant cosine similarity). *Under the setting of Lemma A.7, for any covered token position $t$ and any window $i \in \mathcal{C}(t)$,*

$$\cos\big(q_i', Z_i'(t)\big) = \cos\big(q_i, Z_i(t)\big). \tag{37}$$

*Consequently, the overlap-averaged similarity $\mathrm{sim}(t)$ and the semantic score $\mathrm{Sem}(t)$ in Eq. (6) are unchanged.*

*Proof.* By Lemma A.7, $q_i' = q_i R_i$ and $Z_i'(t) = Z_i(t) R_i$. Because $R_i \in \mathrm{O}(k)$, it preserves inner products and norms: for any row vectors $x, y \in \mathbb{R}^{1 \times k}$,

$$(x R_i)(y R_i)^\top = x R_i (R_i^\top y^\top) = x y^\top, \qquad \|x R_i\|_2 = \|x\|_2. \tag{38}$$

Therefore $\cos(x R_i, y R_i) = \cos(x, y)$, proving the first claim. Overlap averaging and the activation in Eq. (6) depend only on these cosine similarities, so $\mathrm{sim}(t)$ and $\mathrm{Sem}(t)$ remain unchanged. $\square$

**Proposition 4.1 (restated).** Fix the same hyperparameters $(\alpha_{\mathrm{lex}}, \gamma, \lambda_{\mathrm{heat}})$ and consider the evidence score

$$\mathrm{Score}(t) = \mathrm{Base}(t) \cdot \exp(\lambda_{\mathrm{heat}}\, L(t)) \cdot (1 + \xi_{\log}(t))^\gamma, \quad \mathrm{Base}(t) = \mathrm{Sem}(t) + \alpha_{\mathrm{lex}}\mathrm{Lex}(t). \tag{39}$$

Under arbitrary per-window orthogonal basis changes $\{R_i\}_{i=1}^n \subset \mathrm{O}(k)$ (with induced $Z_i', U_{i \leftarrow j}', q_a'$ as in Lemma A.7), we have $\mathrm{Score}'(t) = \mathrm{Score}(t)$ for all positions $t$.

*Proof of Proposition 4.1.* We show invariance term-by-term.

**Semantic term.** By Lemma A.8, $\mathrm{Sem}'(t) = \mathrm{Sem}(t)$.

**Lexical term.** By definition (Eq. (7)), $\text{Lex}(t)$ depends only on token IDs and prompt-local token frequencies, and does not depend on the latent coordinates. Thus $\text{Lex}'(t) = \text{Lex}(t)$ and $\text{Base}'(t) = \text{Base}(t)$.

**Innovation gain.** The innovation term $\xi_{\log}(t)$ is defined from a canonical norm $\|\cdot\|_2$ after mapping into the canonical chart (via $\mathbf{W}_i$). Under a basis change $R_i \in \mathrm{O}(k)$, the canonical coordinates rotate as $\mathbf{W}'_i = \mathbf{W}_i R_i$. Hence for any residual row vector $r$, $\|r\mathbf{W}'_i\|_2 = \|r\mathbf{W}_i R_i\|_2 = \|r\mathbf{W}_i\|_2$ because $R_i$ is orthogonal. Therefore $\xi'_{\log}(t) = \xi_{\log}(t)$ and $(1 + \xi'_{\log}(t))^\gamma = (1 + \xi_{\log}(t))^\gamma$.

**Loss gain.** The loss map $L(t)$ is computed from the backbone hidden states in the backbone's native coordinate system and is independent of any per-window post-hoc chart basis changes in canonical space. Therefore $L'(t) = L(t)$ and $\exp(\lambda_{\text{heat}} L'(t)) = \exp(\lambda_{\text{heat}} L(t))$. In the main results we further set $\lambda_{\text{heat}} = 0$, so this factor is identically 1.

Combining the four parts yields $\text{Score}'(t) = \text{Score}(t)$ for all $t$. $\qquad\square$

### A.6. Loss heatmap bounds (optional diagnostic)

This subsection covers basic mathematical properties of the optional loss heatmap term $L(t)$, which is used only when $\lambda_{\text{heat}} > 0$ (in main results we set $\lambda_{\text{heat}} = 0$).

**Lemma A.9** (Vector-entropy effective rank lies in $[1, D]$). *Let $x \in \mathbb{R}^D$ and define $p_i := |x_i|/\sum_{j=1}^D |x_j|$ (with any convention when the denominator is zero). Define the vector-entropy effective rank $\text{EffRank}(x) := \exp(H(p))$, where $H(p) := -\sum_{i=1}^D p_i \log p_i$ is Shannon entropy. Then*

$$1 \le \text{EffRank}(x) \le D. \tag{40}$$

*Proof.* For any distribution $p$ over $D$ outcomes, $H(p) \in [0, \log D]$. Exponentiating yields $\exp(H(p)) \in [1, D]$. $\qquad\square$

**Lemma A.10** (Computational identity for vector entropy). *Let $x \in \mathbb{R}^D$ and let $S := \sum_{i=1}^D |x_i|$ with $S > 0$. With $p_i = |x_i|/S$, the entropy can be written as*

$$H(p) = \log S - \frac{1}{S}\sum_{i=1}^D |x_i| \log |x_i|. \tag{41}$$

*Proof.* Using $p_i = |x_i|/S$,

$$H(p) = -\sum_i \frac{|x_i|}{S} \log \frac{|x_i|}{S} = -\frac{1}{S}\sum_i |x_i|(\log |x_i| - \log S) = \log S - \frac{1}{S}\sum_i |x_i| \log |x_i|. \tag{42}$$

$\qquad\square$

**Lemma A.11** (Loss heatmap is bounded in $[0, 1]$). *Let $\text{EffRank}(h_t)$ be the per-position effective rank and define the loss map*

$$L(t) := 1 - \text{clip}\left(\frac{\text{EffRank}(h_t)}{\text{EffRank}(h_{\text{ref}})}, 0, 1\right). \tag{43}$$

*Then $L(t) \in [0, 1]$ for all $t$.*

*Proof.* The clipped ratio lies in $[0, 1]$ by definition, so $1 - \text{clip}(\cdot) \in [0, 1]$. $\qquad\square$

### A.7. Holonomy bound (path discrepancy)

**Lemma A.12** (Holonomy bound for two-path transports). *Let $\gamma_1, \gamma_2 : a \to i$ be two paths on the window graph and let their induced path transports be $\Pi_{\gamma_1}, \Pi_{\gamma_2} \in \mathrm{O}(k)$ (products of edge transports). Let the loop be $\mathcal{C} := \gamma_2 \circ \gamma_1^{-1}$ with transport $\Pi_{\mathcal{C}} := \Pi_{\gamma_2}\Pi_{\gamma_1}^{-1}$. Then for any query row vector $q \in \mathbb{R}^{1 \times k}$,*

$$\|q\Pi_{\gamma_2} - q\Pi_{\gamma_1}\|_2 \le \|\Pi_{\mathcal{C}} - I\|_{\text{op}} \|q\|_2. \tag{44}$$

*Proof.* Because $\Pi_{\gamma_2} = \Pi_{\mathcal{C}}\Pi_{\gamma_1}$, we have

$$q\Pi_{\gamma_2} - q\Pi_{\gamma_1} = q(\Pi_{\mathcal{C}} - I)\Pi_{\gamma_1}. \tag{45}$$

Taking norms and using submultiplicativity,

$$\|q(\Pi_{\mathcal{C}} - I)\Pi_{\gamma_1}\|_2 \leq \|q\|_2 \|\Pi_{\mathcal{C}} - I\|_{\mathrm{op}} \|\Pi_{\gamma_1}\|_{\mathrm{op}}. \tag{46}$$

Since $\Pi_{\gamma_1} \in \mathrm{O}(k)$, $\|\Pi_{\gamma_1}\|_{\mathrm{op}} = 1$, giving the claim. $\qquad\square$

**A.8. Proof of Proposition 5.1 (ordered compression feasibility)**

This subsection provides a proof for Proposition 5.1.

**Proposition A.13** (Ordered compression is an order-preserving subsequence)**.** *Let $x_{1:T}$ be an input sequence and let $I \subset \{1, \ldots, T\}$ be any set of retained indices. Define $\tilde{x}$ as the ordered concatenation of retained tokens:*

$$\tilde{x} := (x_t)_{t \in I \text{ sorted increasingly}}. \tag{47}$$

*Then $\tilde{x}$ is an order-preserving subsequence of $x_{1:T}$ (denoted $\tilde{x} \preceq x_{1:T}$), and $|\tilde{x}| = |I|$.*

*Proof.* By construction, $\tilde{x}$ is obtained by selecting tokens from $x_{1:T}$ without changing their relative order, hence it is an order-preserving subsequence. Each index appears at most once in the sorted list, so $|\tilde{x}| = |I|$. $\qquad\square$

**Proposition A.14** (Feasibility of the ordered compression policy)**.** *Let $B$ be the hard budget and let the ordered compression policy output $\tilde{x}$ by selecting a set of indices $I$ and emitting them in increasing order. Then $|\tilde{x}| \leq B$ and $\tilde{x} \preceq x_{1:T}$. Moreover, if $B \geq T$, the policy leaves the sequence unchanged and returns $\tilde{x} = x_{1:T}$.*

*Proof.* Order preservation follows directly from Proposition A.13. It remains to show $|I| \leq B$.

The ordered compression policy keeps a prefix of length $P_{\mathrm{keep}}$ and a tail of length $T_{\mathrm{keep}}$, then allocates at most $B_{\mathrm{mid}} := B - P_{\mathrm{keep}} - T_{\mathrm{keep}}$ indices to the mid region. Thus the total number of kept indices is bounded by

$$|I| \leq P_{\mathrm{keep}} + T_{\mathrm{keep}} + B_{\mathrm{mid}} = B. \tag{48}$$

Finally, if $B \geq T$, there is no budget pressure; the ordered policy keeps all indices, so $\tilde{x} = x_{1:T}$. $\qquad\square$

**Proof of Proposition 5.1.** Proposition 5.1 is the main-text instance of Proposition A.14 above, with $B = \mathtt{max\_len}$ and $T = \mathtt{orig\ length}$.

# B. Background and Notation

This section provides supporting background to make the geometric pipeline easier to audit, along with notation and task references.

## B.1. State Space Models and the Memory Bottleneck

State Space Models (SSMs) such as Mamba (Gu & Dao, 2024) and Mamba-2 (Dao & Gu, 2024) compress sequence history into a fixed-capacity hidden state $h_t \in \mathbb{R}^{d_{\mathrm{model}}}$. While this grants $O(T)$ inference complexity and bounded memory, it also introduces a fundamental bottleneck: information from early tokens may be overwritten before it becomes query-relevant. This *memory bottleneck* manifests as degraded performance on precise-memory tasks (e.g., needle-in-haystack retrieval) when context length grows.

GeoMamba addresses this bottleneck at inference time by selectively *recontextualizing*: rather than relying on the compressed state, we identify and re-inject the most query-relevant evidence into a shorter, order-preserving prompt. The key insight is that under a hard budget, retaining a small set of high-relevance tokens can outperform processing the full lossy context.

## B.2. What is Being Aligned?

GeoMamba treats each overlapping window as a local chart with canonical latents $Z_i \in \mathbb{R}^{w \times k}$ (boundary windows are padded to length $w$ and masked, so only $|W_i| \leq w$ valid tokens contribute). Pairwise transports are estimated on overlap tokens by orthogonal Procrustes, yielding $U_{i \leftarrow j} \in \mathrm{O}(k)$ that approximately preserves inner products in the canonical space. We restrict to $\mathrm{O}(k)$ because cosine similarity is the core semantic signal in covariant retrieval (LAR).

## B.3. Regularized Whitening

Let a window have masked mean $\mu$ and masked covariance $\Sigma \succeq 0$ in the shared $k$-dimensional subspace. With eigendecomposition $\Sigma = Q \operatorname{diag}(\lambda) Q^\top$ and regularization $\lambda_{\mathrm{reg}} > 0$, we whiten with

$$\mathbf{W} = Q \operatorname{diag}\!\big((\lambda + \lambda_{\mathrm{reg}})^{-1/2}\big) Q^\top, \qquad z_{\mathrm{canon}} = (z_{\mathrm{raw}} - \mu)\,\mathbf{W}. \tag{49}$$

**Lemma B.1** (Whitening contracts the spectrum). *The canonical covariance is* $\mathbf{W}^\top \Sigma \mathbf{W} = Q \operatorname{diag}\!\left(\frac{\lambda}{\lambda + \lambda_{\mathrm{reg}}}\right) Q^\top$, *so every eigenvalue lies in* $[0, 1)$.

*Proof.* Substitute and use $Q^\top Q = I$. $\qquad\square$

## B.4. Orthogonal Transports Preserve Similarity

We restrict transports to $\mathrm{O}(k)$ so that cosine similarity is invariant under transport.

**Lemma B.2** (Cosine invariance). *For any* $x, y \in \mathbb{R}^{1 \times k}$ *and* $U \in \mathrm{O}(k)$: $\cos(xU, yU) = \cos(x, y)$.

## B.5. Notation Reference

To keep the appendix compact, we list only *core symbols reused across sections* in Table 3. Symbols that appear only inside a single equation (e.g., intermediate statistics inside Eqs. (7)–(9)) are defined locally at first use. We use 1-based indexing for token positions ($t \in \{1, \ldots, T\}$) to match the main text.

*Table 3.* Core notation reference.

| Symbol | Meaning | Symbol | Meaning |
|---|---|---|---|
| *Sequence, windows, and cover graph* | | | |
| $x_{1:T}$ | Input token sequence | $T$ | Sequence length |
| $x_t$ | Token at position $t$ (ID) | $e_t$ | Input embedding of token $x_t$ |
| $t$ | Absolute token position (1-based) | $w,\, s$ | Window size and stride |
| $a_i,\, W_i$ | Window start index and token index set | $n$ | Number of windows |
| $\Omega_{ij}$ | Overlap set $W_i \cap W_j$ | $\mathcal{C}(t)$ | Windows covering position $t$ |
| $G = (V, E, \omega)$ | Window overlap graph | $\omega_{ij}$ | Overlap weight |
| $U_{i \leftarrow j}$ | Orthogonal transport (chart $j \to i$) | $\Gamma_{ij}$ | Overlap weights in Procrustes |
| | | $p,\, g_{\min}$ | |
| $\tau$ | Overlap threshold | $k_{\mathrm{virt}},\, \eta_{\mathrm{virt}}$ | Virtual-edge params (optional) |
| *Local charts and geometry* | | | |
| $d_{\mathrm{model}}$ | Backbone hidden dimension | $k$ | Shared latent dimension |
| $H_i^{\mathrm{bb}}$ | Backbone states for window $W_i$ | $Z_i^{\mathrm{raw}},\, Z_i$ | Raw / canonical latents |
| $\mathbf{W}_i$ | Whitening map | $\rho$ | Whitening regularization ratio |
| $P_{\mathrm{ws}}$ | Warm-start prefix length | $v_i(t)$ | Validity mask in window $i$ |
| $b$ | Burn-in length (masked prefix per window) | $\mu_i$ | Masked mean in window $i$ |
| *Retrieval, scoring, and compression* | | | |
| $a$ | Anchor (tail) window index | $q$ | Tail query length |
| $\mathrm{Tail}(a)$ | Last $q$ valid tail indices in window $a$ | $q_a,\, q_i$ | Anchor / transported query |
| $\pi(a \to i)$ | Best path from $a$ to window $i$ | $\mathrm{sim}(t)$ | Overlap-averaged similarity |
| $\beta$ | Similarity sharpness | $\mathrm{Sem}(t)$ | Semantic score |
| $\mathrm{Lex}(t)$ | Lexical overlap score | $\alpha_{\mathrm{lex}}$ | Lexical weight |
| $f_{\max}$ | Lexical frequency cap | $\mathrm{idf}(u)$ | Prompt-local IDF proxy |
| $\gamma$ | Innovation exponent | $\mathrm{Score}(t)$ | Final evidence score |
| $K,\, r$ | Seeds and chunk radius | $m(t)$ | Keep mask |
| $B$ | Token budget | $B_{\mathrm{mid}}$ | Remaining mid budget |
| $P_{\mathrm{keep}},\, T_{\mathrm{keep}}$ | Prefix/tail tokens always kept | $\tilde{x}$ | Compressed prompt (order-preserving) |

**B.6. Task Name Abbreviations**

**RULER NiAH (FULL8).**

- **S1**: Single-needle retrieval (depth 1).

- **S2**: Single-needle retrieval (depth 2).

- **S3**: Single-needle retrieval (depth 3).

- **MK1**: Multi-key binding (1 key).

- **MK2**: Multi-key binding (2 keys).

- **MK3**: Multi-key binding (3 keys).

- **MV**: Multi-value retrieval.

- **MQ**: Multi-query retrieval.

**LongBench-E (e-13).**

- **PC**: Passage count (counting).

- **PR**: Passage retrieval (retrieval).

- **GR**: GovReport (summarization).

- **MN**: MultiNews (summarization).

- **MQA**: MultiFieldQA (single-doc QA).

- **QA**: Qasper (single-doc QA).

- **2WM**: 2WikiMQA (multi-doc QA).

- **HQA**: HotpotQA (multi-doc QA).

- **SS**: SAMSum (summarization).

- **TR**: TREC (few-shot classification).

- **TQA**: TriviaQA (single-doc QA).

- **LCC**: LCC (code completion).

- **RB**: RepoBench-P (code completion).

**B.7. Useful Diagnostics**

Two diagnostics are particularly informative in practice:

- **Overlap alignment residual:** for edge $(i, j)$, measure $r_{ij} = \frac{1}{\sqrt{|\Omega_{ij}|}} \|Z_i[\Omega_{ij}] - Z_j[\Omega_{ij}] U_{i \leftarrow j}\|_F$. High residuals indicate poor geometric alignment.

- **Path reliability:** best-path transport prefers paths with larger overlap-weight products. Unusually long or low-reliability paths signal weak overlap connectivity.

**B.8. Implementation Notes**

This subsection provides algorithmic details and evaluation protocols sufficient for reproduction.

**End-to-end algorithm.** Algorithm 2 provides detailed pseudocode for the full GeoMamba pipeline. All steps are performed at inference time with a frozen backbone; no additional parameters are trained.

---

**Algorithm 2** GeoMamba: Geometry-Guided Recontextualization (Detailed)

---

**Require:** Input tokens $x_{1:T}$, frozen backbone $\mathcal{M}$, budget $B$, hyperparameters (Table 4)
**Ensure:** Compressed prompt $\tilde{x}$ (order-preserving subsequence of $x_{1:T}$)
 1: **Cover construction:** Partition $x_{1:T}$ into overlapping windows $\{W_i\}_{i=1}^n$ with size $w$ and stride $s$
 2: **Window-independent encoding:** For each $W_i$, run $\mathcal{M}$ from a fresh state to obtain hidden states $H_i^{\text{bb}}$
 3: **Shared subspace selection:** Select $k$ backbone dimensions with highest conservative masked variance across windows (min-over-windows; Eq. (18))
 4: **Whitening:** For each $W_i$, estimate masked mean/covariance and apply regularized whitening to obtain canonical latents $Z_i$
 5: **Overlap transports:** For each edge $(i,j)$ in the cover graph, measure $U_{i\leftarrow j} \in O(k)$ by weighted Procrustes on overlap tokens
 6: **Best-path transport:** Compute shortest paths (on $-\log\omega$ weights) from the anchor window $a$ to all windows
 7: **Query construction:** Build anchor query $q_a$ from the last $q$ tokens of window $a$ (IDF-weighted pooling; optional multi-query); transport to each window via best-path composition
 8: **Evidence scoring:** Compute $\text{Score}(t)$ via LAR (semantic+lexical with optional gains; Eqs. (8)–(10))
 9: **Seed selection:** Select top-$K$ span seeds by energy (span length $\ell$; Eq. (12)) and expand to chunks of radius $r$
10: **Ordered compression:** Keep prefix/tail tokens; fill remaining budget with highest-scoring chunks; emit in original order
11: **Generation:** Run $\mathcal{M}$ on $\tilde{x}$ to produce the output

---

*Table 4.* **GeoMamba hyperparameters (main runs).** Shared geometry: $(w, s, k) = (512, 192, 128)$, overlap threshold $\tau = 0.1$, warm-start $P_{\text{ws}} = 0$ (0 disables), burn-in $b = 4$, similarity sharpness $\beta = 8$ (sigmoid), whitening regularization $\rho = 10^{-3}$, lexical cap $f_{\max} = 128$, virtual edges ($top_p = 0.05$, $g_{\min} = 3$, $k_{\text{virt}} = 2$, $\eta_{\text{virt}} = 0.5$), and Stage-7 scaffolds (fixed stride=64, keep=4) with fill-to-budget enabled.

| Group | RULER NiAH (ctx=4096→2048) | LongBench-E (ctx=16384→4096) |
|---|---|---|
| Budget / keeps | $B$=2048
prefix$_{\text{keep}}$=0
tail$_{\text{keep}}$=256
$q$=256 | $B$=4096
prefix$_{\text{keep}}$=384
tail$_{\text{keep}}$=512
$q$=256 |
| Retrieval | $K$=256
$r$=16
loss (power) $= 0$ | $K$=256
$r$=16
loss (power) $= 0$ |
| Gains | lex $\alpha_{\text{lex}}$=1
innov $\gamma$=1 | lex $\alpha_{\text{lex}}$=1.0
innov $\gamma$=1 |

B.8.1. WINDOW COVER

We build a fixed-size overlapping window cover and overlap graph. Given a token length $T$, window size $w$, and stride $s < w$, windows are $W_i = [a_i, a_i + w - 1]$ with $a_i = 1 + (i-1)s$. We also support:

- **Burn-in masking** ($b$): the first few tokens of each window are excluded from geometry statistics to reduce fresh-state artifacts.

- **Attention masks**: all window statistics and overlaps are masked to ignore padding.

B.8.2. WINDOW-INDEPENDENT ENCODING AND WARM-START PREFIX

Each window is encoded from a *fresh* backbone state (window-independent, WI) to avoid a single streaming bottleneck. For recurrent backbones, fresh-state transients can corrupt early token representations. We mitigate this with an optional

**warm-start prefix**: for each window $W_i = [a_i, a_i + w - 1]$, we prepend a short left-context prefix of length $P_{\text{ws}}$ consisting of tokens immediately preceding $a_i$ (clipped to the sequence start). The backbone is run on the concatenated sequence

$$(x_{a_i - P_{\text{ws}} : a_i - 1}, \ x_{a_i : a_i + w - 1}), \tag{50}$$

but we only keep hidden states aligned to the window tokens $x_{a_i : a_i + w - 1}$. This preserves WI semantics (the window is still decoded from a reset state) while reducing boundary drift. Main runs set $P_{\text{ws}} = 0$ (0 disables warm-start).

### B.8.3. SHARED SUBSPACE SELECTION (VARIANCE TOP-$k$)

To keep transport coordinates stable across windows, we select a single shared subset of $k$ backbone dimensions by masked variance. We rank dimensions by the minimum per-window variance (Eq. (18)) to avoid coordinates that are nearly constant in any window, which can destabilize whitening and overlap Procrustes. Unlike per-window PCA, this avoids basis "rotation" that would otherwise leak into the transport estimates.

### B.8.4. GAUGE–METRIC CALIBRATION (WHITENING) AND TRANSPORTS

We separate **metric** from **gauge**. For each window, we estimate a masked mean/covariance in the shared $k$-dimensional subspace and apply regularized whitening. We then measure orthogonal overlap transports by weighted Procrustes in canonical coordinates.

**Virtual edges (optional).** In addition to adjacent overlap edges, we optionally add non-adjacent *virtual edges* between semantically similar windows. Virtual edges provide shortcut paths that reduce error accumulation when composing many adjacent transports, without introducing any global frame estimation. We compute a mean-pooled window embedding $\bar{z}_i = \frac{\sum_t v_i(t) Z_i(t)}{\sum_t v_i(t)}$ and pairwise cosine similarities $\kappa_{ij} = \cos(\bar{z}_i, \bar{z}_j)$. We keep only the top-$p$ fraction of eligible non-adjacent similarities (pairs with $|i - j| \geq g_{\min}$), and retain at most $k_{\text{virt}}$ outgoing virtual edges per window. Virtual-edge reliabilities are $\omega_{ij}^{\text{virt}} = \min(1, \eta_{\text{virt}} \cdot \max(0, \kappa_{ij})) \in (0, 1]$. Default hyperparameters: $(p, g_{\min}, k_{\text{virt}}, \eta_{\text{virt}})$ are reported in Table 4.

### B.8.5. COVARIANT RETRIEVAL (LAR)

It computes a per-position score combining semantic similarity and saliency:

$$\text{Score}(t) = \text{Base}(t) \cdot \exp(\lambda_{\text{heat}} \, L(t)) \cdot (1 + \xi_{\log}(t))^{\gamma}, \tag{51}$$

using:

- **Content-tail query**: the query is built from the last $q$ tokens of the prompt using prompt-local IDF-weighted pooling (with an optional multi-query variant built from top-IDF tail tokens).

- **Path-covariant similarity**: the query is transported across windows by best-path transport (path composition of measured $U_{i \leftarrow j}$), and cosine similarity is computed in each local chart.

- **Lexical overlap (optional)**: a task-agnostic token-ID overlap term can be added into $\text{Base}(t) = \text{Sem}(t) + \alpha_{\text{lex}} \text{Lex}(t)$.

- **Loss heatmap** (optional): when enabled ($\lambda_{\text{heat}} > 0$), we compute an effective-rank deficit map and apply a multiplicative gain $\exp(\lambda_{\text{heat}} L(t))$.

- **Innovation saliency**: we compute an affine-connection residual magnitude $\xi_{\log}(t)$ and apply $(1 + \xi_{\log}(t))^{\gamma}$ as a gain.

The retrieval block also provides lightweight diagnostics such as the effective query length and basic score statistics.

### B.8.6. ORDERED CHUNK COMPRESSION UNDER A HARD BUDGET

Ordered compression constructs a budgeted prompt $\tilde{x}$ that is an *ordered subsequence* of the original prompt. The ordered compression policy is:

- optionally keep a fixed prefix (system/prompt header) and keep a fixed tail (question/instruction),

- convert retrieved seeds (span- or token-based) into contiguous chunks (seed expansion),

- select whole chunks under the remaining budget, ranked by the sum of per-token scores within each chunk,

- emit the final prompt by **ordered** concatenation (no token reordering).

B.8.7. REPRODUCING BENCHMARK RUNS

Our evaluation pipeline reproduces both full-context baselines and GeoMamba recontextualization runs.

**Official prompts and scorers.**  We use the benchmark-provided prompts and official scorers for RULER and LONGBENCH (wrapped in our evaluation adapters).

**Tokenization and context-length clipping.**  When using an instruct/chat backbone with an available chat template, we encode prompts via the chat template; otherwise we encode raw text. For RULER, answer cues live at the prompt tail, so when enforcing a fixed context length we clip from the left (keep the suffix). For LONGBENCH, prompts can exceed the context length, so we apply head+tail clipping that keeps the first half and the last half of the prompt (the question typically lives at the tail).

**Decoding.**  We use greedy decoding and follow the benchmark-provided generation-length limits (including any per-task limits for LongBench). For LongBench-E, per-task max_new_tokens are fixed by the official benchmark configuration and shared across all methods; we do not tune them.

**Budget and compression.**  In our notation, the hard budget $B$ is the maximum number of tokens kept in the recontextualized prompt. We reserve fixed prefix/tail portions (system prompts and question cues) before allocating the remaining budget to retrieved evidence.

# C. Additional Experiments

This appendix provides additional evidence on hyperparameter selection and results on Mamba2-130M (with an external DeciMamba reference). Each subsection addresses a specific question raised by the main results and ends with a takeaway.

## C.1. Hyperparameter Selection Protocol

**Question.**  How are GeoMamba hyperparameters selected, and is there per-task tuning?

**Protocol.**  We select a *single* GeoMamba configuration per benchmark on a fixed calibration subset and then freeze it for all reported runs. For NiAH we tune on the full 8-task suite ($n$=200 samples/task); for LongBench-E we tune on a 4-task representative subset (Qasper, HotpotQA, PassageRetrieval-en, RepoBench-P) with $n$=200 total examples to keep calibration compute manageable, and then freeze the chosen setting for the full e-13 evaluation. The calibration subset is held out from the reported evaluation split. Table 5 summarizes the selected configuration.

*Table 5.* **GeoMamba hyperparameter selection (benchmark-level).** A single configuration is selected per benchmark on a fixed calibration subset and then frozen for all reported runs (no per-task tuning). For NiAH, $n$ is per-task; for LongBench-E, $n$ is the total number of calibration examples across the 4-task tuning subset.

| Benchmark | ctx | Budget $B$ | Calib. $n$ | $\alpha_{\text{lex}}$ | $K$ (topk) | max_retrieved |
|---|---|---|---|---|---|---|
| RULER NiAH (FULL8) | 4096 | 2048 | 200 | 1.0 | 256 | 256 |
| LongBench-E (e-13) | 16384 | 4096 | 200 | 1.0 | 256 | 512 |

**Takeaway.**  On the calibration subset, the best and runner-up settings are close (NiAH: 25.1 vs. 25.0; LongBench-E tuning subset: 11.5 vs. 11.5), indicating low sensitivity to exact hyperparameter values once the order of magnitude is correct.

## C.2. Results on Mamba2-130M

**Question.**  Do GeoMamba's gains persist on a smaller backbone?

**Setup.** We report full-context Mamba2-130M (Vanilla) and GeoMamba under the same generation budgets as the main text. For reference, we also include DeciMamba using the official DeciMamba-130M checkpoint (Mamba-1-based); because the backbone differs from Mamba2, these numbers are not directly comparable and serve as external context only. Tables 6 and 7 summarize NiAH FULL8 and LongBench-E (e-13), respectively.

*Table 6.* **RULER NiAH (FULL8, ctx=4096, $n$=500) on Mamba2-130M.** Task accuracy (%) on eight NiAH variants. We report full-context Mamba2-130M (Vanilla), DeciMamba-130M (official; Mamba-1-based external baseline (Ben-Kish et al., 2024)), and GeoMamba on frozen Mamba2-130M using a 50% *generation* budget ($B$=2048). *Takeaway:* GeoMamba improves average accuracy under a strict generation budget.

| Model | Method | S1 | S2 | S3 | MK1 | MK2 | MK3 | MV | MQ | Avg. |
|---|---|---|---|---|---|---|---|---|---|---|
| Mamba2-130M | Vanilla | 99.8 | 26.4 | 0.8 | 15.2 | 0.0 | 0.0 | 0.1 | 0.0 | 17.8 |
| Mamba-130M | DeciMamba | 90.8 | 39.0 | 1.8 | 15.0 | 0.0 | 0.0 | 5.8 | 0.1 | 19.1 |
| Mamba2-130M | **GeoMamba** | **98.8** | **59.6** | **7.2** | **26.2** | **0.0** | **0.0** | **2.7** | **0.9** | **24.4** |

*Table 7.* **LongBench-E (e-13, ctx=16384, $n$=300) on Mamba2-130M.** Scores on the 13-task English subset. We report full-context Mamba2-130M (Vanilla), DeciMamba-130M (official; Mamba-1-based external baseline (Ben-Kish et al., 2024)), and GeoMamba on frozen Mamba2-130M using a *generation* budget $B$=4096. *Takeaway:* GeoMamba improves overall performance under a strict generation budget, with gains concentrated in code and classification.

| Model | Method | Synthetic | | Summary | | Single-doc QA | | Multi-doc QA | | Few-shot Learning | | | Coding | | Avg. |
|---|---|---|---|---|---|---|---|---|---|---|---|---|---|---|---|
| | | PC | PR | GR | MN | MQA | QA | 2WM | HQA | SS | TR | TQA | LCC | RB | |
| Mamba2-130M | Vanilla | 2.9 | 0.5 | 6.7 | 10.1 | 9.1 | 3.1 | 7.8 | 4.6 | 3.7 | 26.7 | 17.4 | 31.8 | 33.5 | 12.15 |
| Mamba-130M | DeciMamba | 3.7 | 1.3 | 0.0 | 0.0 | 2.2 | 4.5 | 6.1 | 3.3 | 0.5 | 4.7 | 9.2 | 29.1 | 26.5 | 7.0 |
| Mamba2-130M | **GeoMamba** | **3.1** | **0.7** | **7.6** | **10.7** | **10.9** | **3.8** | **8.5** | **5.3** | **4.7** | **37.7** | **19.0** | **39.0** | **35.2** | **14.32** |

**Geometry ablations (Mamba2-130M, LongBench-E).** Table 8 ablates the geometry pipeline under the same budget. Removing transports, whitening, or best-path query transport reduces the average score by about 3–4%, indicating that gains are not attributable to a single tweak but to the full geometric calibration stack.

*Table 8.* **Geometry ablations on LongBench-E (e-13) for GeoMamba on Mamba2-130M.** Scores on the 13-task English subset under a fixed *generation* budget $B$=4096. We ablate (i) overlap transports (**–Transport**), (ii) metric calibration (**–Whitening**), and (iii) best-path query transport (**–Bestpath**). *Takeaway:* Each component contributes about 3–4% on average.

| Task | Full | –Transport | –Whitening | –Bestpath |
|---|---|---|---|---|
| qasper_e | 3.8 | 3.5 | 3.3 | 3.2 |
| multifieldqa_en_e | 10.9 | 10.6 | 10.6 | 10.8 |
| hotpotqa_e | 5.3 | 4.9 | 4.9 | 5.0 |
| 2wikimqa_e | 8.5 | 7.8 | 8.1 | 7.8 |
| gov_report_e | 7.6 | 7.7 | 7.6 | 7.5 |
| multi_news_e | 10.7 | 10.6 | 10.6 | 10.7 |
| trec_e | 37.7 | 33.7 | 34.3 | 34.7 |
| triviaqa_e | 19.0 | 17.8 | 18.4 | 18.6 |
| samsum_e | 4.7 | 4.6 | 4.6 | 4.7 |
| passage_retrieval_en_e | 0.7 | 0.6 | 0.7 | 0.5 |
| passage_count_e | 3.1 | 3.2 | 3.3 | 2.8 |
| lcc_e | 39.0 | 38.5 | 38.6 | 37.5 |
| repobench-p_e | 35.2 | 35.1 | 35.1 | 34.8 |
| **Avg.** | **14.32** | **13.74 (-4.1%)** | **13.85 (-3.3%)** | **13.73 (-4.1%)** |

**Component ablations (Mamba2-130M, LongBench-E).** Table 9 ablates auxiliary scoring components: lexical overlap, innovation residual, virtual edges, and scaffold anchors. Each component contributes modestly (1–4%), with virtual edges showing the largest individual effect. These ablations are orthogonal to the geometry ablations above: the geometry stack (Table 8) provides the core calibration, while these components provide task-agnostic refinements.

*Table 9.* **Component ablations on LongBench-E (e-13) for GeoMamba on Mamba2-130M.** Scores on the 13-task English subset under a fixed *generation* budget $B$=4096. We ablate auxiliary scoring components: (i) lexical overlap (**–Lexical**), (ii) innovation residual (**–Innovation**), (iii) virtual edges (**–Virtual**), and (iv) scaffold anchors (**–Scaffold**). *Takeaway:* Each component contributes modestly; the core geometry (Table 8) accounts for the largest gains.

| Task | Full | –Lexical | –Innovation | –Virtual | –Scaffold |
|---|---|---|---|---|---|
| qasper_e | 3.80 | 3.20 | 3.52 | 3.19 | 3.24 |
| multifieldqa_en_e | 10.90 | 10.54 | 10.40 | 10.80 | 10.28 |
| hotpotqa_e | 5.30 | 4.63 | 4.99 | 4.97 | 4.84 |
| 2wikimqa_e | 8.50 | 7.81 | 7.63 | 7.75 | 7.74 |
| gov_report_e | 7.60 | 7.44 | 7.61 | 7.49 | 7.95 |
| multi_news_e | 10.70 | 10.46 | 10.67 | 10.68 | 10.80 |
| trec_e | 37.70 | 36.67 | 33.67 | 34.00 | 37.67 |
| triviaqa_e | 19.00 | 18.02 | 18.44 | 18.57 | 18.60 |
| samsum_e | 4.70 | 4.27 | 4.54 | 4.74 | 4.86 |
| passage_retrieval_en_e | 0.70 | 0.56 | 0.61 | 0.50 | 0.62 |
| passage_count_e | 3.10 | 3.23 | 3.32 | 3.01 | 3.70 |
| lcc_e | 39.00 | 38.27 | 39.77 | 37.64 | 38.25 |
| repobench-p_e | 35.20 | 35.71 | 35.35 | 34.80 | 35.04 |
| **Avg.** | **14.32** | 13.91 (-2.9%) | 13.89 (-3.0%) | 13.70 (-4.3%) | 14.12 (-1.4%) |

**Budget sensitivity (Mamba2-130M, LongBench-E).** Table 10 varies the *generation* budget $B$ on four representative LongBench-E tasks. Performance is not monotonic in $B$; for example, passage retrieval improves at $B$=2048 (1.29) relative to $B$=4096 (0.70), consistent with the idea that additional mid-context tokens can reintroduce distractors under a recurrent memory bottleneck.

*Table 10.* **Budget sensitivity on LongBench-E (e-13) for GeoMamba on Mamba2-130M.** Scores on four representative tasks when varying the *generation* budget $B$. Mamba2 is the full-context baseline; GeoMamba generates from an order-preserving compressed prompt under budget $B$. *Takeaway:* Performance can be non-monotonic in $B$, especially on retrieval.

| Task | Mamba2 | $B$=2048 | $B$=4096 | $B$=8192 |
|---|---|---|---|---|
| hotpotqa_e | 4.60 | 5.45 | 5.30 | 4.94 |
| passage_retrieval_en_e | 0.50 | 1.29 | 0.70 | 0.89 |
| qasper_e | 3.10 | 3.63 | 3.80 | 3.30 |
| 2wikimqa_e | 7.80 | 8.68 | 8.50 | 8.21 |

**Takeaway.** On Mamba2-130M, GeoMamba improves NiAH FULL8 average from 17.8 to 24.4 and LongBench-E average from 12.15 to 14.32; geometry ablations and budget sensitivity support a causal story where calibrated cross-window similarity and careful budgeting drive the gains.

## C.3. Preprocessing Overhead

Table 11 reports wall-clock timing on LongBench-E for Mamba2-1.3B. GeoMamba's preprocessing adds ∼1.5s per sample on average, dominated by window-independent encoding ($O(n)$ backbone forward passes on length-$w$ windows). The geometry estimation itself (whitening, Procrustes, scoring) is negligible (∼18ms). This overhead is parallelizable across windows; for latency-sensitive applications, it trades compute for improved recall under strict budgets.

*Table 11.* **Preprocessing overhead on LongBench-E (e-13) for GeoMamba on Mamba2-1.3B.** Wall-clock time (seconds per sample). GeoMamba's overhead comes from window-independent encoding; the geometry computation itself is negligible (∼18ms). This preprocessing is parallelizable across windows.

| Task | Baseline | GeoMamba | Overhead |
|------|----------|----------|----------|
| qasper_e | 1.51s | 3.23s | +1.72s |
| multifieldqa_en_e | 0.62s | 1.67s | +1.05s |
| hotpotqa_e | 0.35s | 1.21s | +0.86s |
| 2wikimqa_e | 0.35s | 1.21s | +0.87s |
| gov_report_e | 5.03s | 10.17s | +5.14s |
| multi_news_e | 5.03s | 9.66s | +4.62s |
| trec_e | 0.66s | 1.43s | +0.77s |
| triviaqa_e | 0.35s | 0.88s | +0.54s |
| samsum_e | 0.62s | 1.49s | +0.87s |
| passage_retrieval_en_e | 0.35s | 0.92s | +0.57s |
| passage_count_e | 0.35s | 0.89s | +0.54s |
| lcc_e | 0.62s | 1.34s | +0.72s |
| repobench-p_e | 0.67s | 1.72s | +1.05s |
| **Avg.** | **1.27s** | **2.76s** | **+1.49s** |

## C.4. Limitations and Failure Modes

This subsection documents known limitations and failure modes to guide future work and practical deployment.

**Multi-key binding tasks (MK2/MK3).** GeoMamba does not solve multi-key binding tasks (MK2/MK3) at either the 130M or 1.3B scale; scores remain near zero. These tasks require binding multiple key–value pairs simultaneously, which may exceed the capacity of single-pass recontextualization under a hard budget. A potential remedy is iterative recontextualization: after a first pass, use model-generated partial answers as additional query context and re-score evidence. We leave this extension for future work.

**Preprocessing overhead.** GeoMamba trades preprocessing compute (window-independent encoding) for improved recall under strict budgets. The overhead is dominated by $O(n)$ backbone forward passes on length-$w$ windows. For latency-sensitive applications, this additional compute may be undesirable; the preprocessing is parallelizable across windows.

**Dependence on backbone hidden states.** GeoMamba relies on the quality of backbone hidden states to define local charts and transports. If the backbone exhibits severe hidden-state collapse or degenerate rank, the geometric pipeline may produce low-quality transports. We mitigate this with regularized whitening and stable-dimension selection, but extreme backbone pathologies remain outside the scope of this work.

**Fixed budget vs. adaptive selection.** The current pipeline uses a fixed hard budget $B$ for all prompts. In practice, different prompts may benefit from different compression ratios. Extending GeoMamba to adaptive budgets (e.g., based on holonomy diagnostics or score entropy) is a natural direction for future work.

**Order preservation constraint.** By design, GeoMamba produces an order-preserving subsequence of the original prompt. This constraint ensures compatibility with causal language models but may discard useful information when reordering could improve coherence. For non-causal or bidirectional models, relaxing this constraint could yield further gains.

**Tail-pooled query assumption.** GeoMamba constructs the query from the last $q$ tokens of the prompt, assuming the question or instruction resides at the tail. This assumption holds for RULER and LongBench-E formats but may underperform when relevant cues appear earlier or when multi-turn reasoning is required. Extending to multi-anchor queries (e.g., combining head and tail) is a natural direction.

**Cross-window dependencies.** Window-independent encoding resets the recurrent state for each window, potentially discarding long-range dependencies that span multiple windows. The pipeline relies on overlap transports to recover

cross-window coherence; performance may degrade with small overlaps or highly nonlocal dependencies not captured by adjacent-window geometry.

**Hyperparameter complexity.** The scoring pipeline involves multiple hyperparameters (windowing, stride, $k$, whitening regularization, $\beta$, $\alpha_{\text{lex}}$, $\gamma$, $K$, $r$, virtual edges). While we find low sensitivity to exact values (Section C.1), transferring configurations across domains or backbone families may require recalibration.

## C.5. Reproducibility Statement

All experiments use publicly available model checkpoints (Mamba-2 130M and 1.3B) and benchmark datasets (RULER, LongBench-E). Hyperparameters are reported in Table 4 and frozen across all runs within each benchmark. We use greedy decoding and official scorers for all evaluations. Theoretical claims are supported by self-contained proofs in Appendix A. Upon acceptance, we will release the complete codebase and evaluation scripts.

