# OpenReview forum: "GeoMamba: Geometry-Guided Recontextualization for Precise Long-Context Memory"
_ICML.cc/2026/Conference — Submitted to ICML 2026_

### Official Review · Reviewer_NpP4 · 2026-03-07

**Soundness:** 2
**Presentation:** 2
**Significance:** 2
**Originality:** 2
**Overall Recommendation:** 3
**Confidence:** 3

**Summary:**

This paper addresses three strict settings: frozen SSM backbones, a hard input budget of B<T (applied only to the final inference stage), and order-preserving recontextualization. Its core insight is that selecting tokens with high evidential value and generating in causal order via geometrically calibrated cross-window similarity measurement can achieve a higher recall rate than streaming the entire context through the recurrent backbone. The method is experimentally validated on the Mamba2-1.3B model. It improves the average accuracy from 13.6% to 26.0% on the RULER NiAH FULL8 task (context length 4096 → input budget 2048), and the score from 8.26 to 19.24 on the LongBench-E task (context length 16384 → input budget 4096).

**Compliance With Llm Reviewing Policy:**

Affirmed.

**Final Justification:**

I keep my score for this paper.

**Key Questions For Authors:**

1 No ablation study results are presented in the main text, and the core ablation results for geometric components and auxiliary components are all placed in the appendix. Why did you not select and visualize the key ablation results for analysis in the main text? Have you considered adding charts related to core ablation results in the main text?

2 The experiments only validate two context lengths (4096 and 16384). Can the performance of GeoMamba remain stable at longer context lengths?

3 Table 10 in the appendix shows that on tasks such as HotpotQA, the performance at B=2048 is actually higher than that at B=4096. This phenomenon is explained as "additional mid-context tokens may reintroduce distractions". Does this imply a systemic flaw in GeoMamba's evidence selection mechanism—that is, high-scoring tokens are not necessarily the evidence truly required by the task, and increasing the input budget only introduces noise instead?

4 The method is currently only validated on the Mamba2-130M and 1.3B models. How does this method perform on other mainstream SSM models or hybrid SSM-attention models? Is it necessary to adjust hyperparameters or modify the method for the hidden state characteristics of different models?

**Limitations:**

See the weakness. I hope the authors wll adress my concerns in the rebuttal, My final score depends on the authors' response

**Strengths And Weaknesses:**

### Strengths

1 It has a solid theoretical foundation. The appendix provides self-consistent and rigorous mathematical proofs, with all proof steps elaborated in detail.

2 The method design is reasonable. The authors adopt an objective experimental attitude, analyze the limitations of the method in detail in the appendix, and quantify metrics related to practical deployment such as preprocessing overhead and budget sensitivity.

3 The paper is clearly written with a coherent and logical overall structure.

### Weaknesses

1 The figures are unclear and severely insufficient. There is no visualization of core geometric calibration processes (e.g., window representation alignment, best-path transport) and key experimental results (e.g., ablation studies, performance comparison across tasks), making it hard to intuitively understand the core mechanism of the method.

2 Key experimental results are placed in the appendix. All core experimental contents including ablation studies, small model validation and budget sensitivity analysis are only in the appendix, which impairs the understanding of the method's completeness.

3 There is insufficient plain explanation of geometric concepts. Some terms in the paper are overly professional and not fully illustrated.

4 The experiments are only validated on the Mamba2 series models (130M, 1.3B), with no tests on other SSM architectures or hybrid architectures. This limits the application scope of the method to a certain extent, and the absolute performance of the model remains relatively low.

---

> ### Author Rebuttal · Authors · 2026-03-28
>
> We do appreciate your constructive comments and provide our responses as follows.
>
> # W1/ W2/ Q1. Main-text Placement and Visualization of Core Results
>
> Appendix C.2 already contains the Mamba2-130M validation results in Tables 6-7, the geometry ablations in Table 8, the auxiliary-component ablations in Table 9, and the budget sensitivity analysis in Table 10. These belong to the main empirical story rather than appendix-only support.
>
> Following this suggestion, we will move a condensed block of Tables 6-10 into the main text and summarize the main patterns. We will also add a compact ablation figure based on Tables 8-10, so that the effects of the geometry core, the auxiliary components, and the budget choice can be read directly in the paper body.
>
> # W3. Plain-language Mechanism and Terminology
>
> In a nutshell, GeoMamba builds local window representations, aligns neighboring windows through overlaps, transports the query across windows, scores evidence, compresses the prompt in original order, and lets the frozen backbone answer from the compressed prompt. The main logic chain is:
>
> Window-local encoding -> overlap-based alignment -> query transport -> evidence scoring -> order-preserving compression -> final generation.
>
> Window-local encoding builds local representations without one long recurrent pass. Overlap-based alignment uses shared content to align neighboring windows. Query transport carries the query through that chain. Evidence scoring identifies relevant spans after transport. Order-preserving compression keeps useful evidence in source order under the final budget. Final generation answers from the compressed prompt.
>
> In the revision, we will move this logic chain to the front of the method section and rewrite the main terms more directly. Whitening normalizes local window representations so that cross-window scores are comparable. Overlap transport connects neighboring coordinate systems through shared regions. Best-path query transport routes the query through the most reliable overlap chain. Order-preserving compression keeps useful evidence without changing causal order.
>
> # Q2. Stability at Longer Context Lengths
>
> Following this question, we evaluate GeoMamba under longer contexts in the same FULL8 task space. The results are listed below, with 100 sequences for each task.
>
> | Method | Context | S1 | S2 | S3 | MK1 | MK2 | MK3 | MV | MQ | Avg |
> | --- | ---: | ---: | ---: | ---: | ---: | ---: | ---: | ---: | ---: | ---: |
> | Mamba2 | 8192 | 100.0 | 1.0 | 0.0 | 0.5 | 0.0 | 0.0 | 0.0 | 0.0 | 12.7 |
> | GeoMamba | 8192 | 100.0 | 19.0 | 4.0 | 15.0 | 0.0 | 0.0 | 10.0 | 9.0 | 19.6 |
> | Mamba2 | 16384 | 0.5 | 0.0 | 0.0 | 0.0 | 0.0 | 0.0 | 0.0 | 0.0 | 0.1 |
> | GeoMamba | 16384 | 51.0 | 6.0 | 3.0 | 8.0 | 0.0 | 0.0 | 4.0 | 4.0 | 9.5 |
> | Mamba2 | 32768 | 0.0 | 0.0 | 0.0 | 0.0 | 0.0 | 0.0 | 0.0 | 0.0 | 0.0 |
> | GeoMamba | 32768 | 7.0 | 0.0 | 0.0 | 1.0 | 0.0 | 0.0 | 0.0 | 0.0 | 1.0 |
>
> We observe the following. GeoMamba retains stronger long-context signal than Mamba2 as context grows. At 8192, the average score improves from 12.7 to 19.6. At 16384, where Mamba2 nearly collapses, GeoMamba still improves the average score from 0.1 to 9.5, with a clear gain on S1 (51.0 vs. 0.5). At 32768, GeoMamba still retains a small residual advantage. In summary, GeoMamba degrades more gracefully than vanilla Mamba2 under longer contexts.
>
> # Q3. Why B=2048 Can Outperform B=4096 on Some Tasks
>
> Table10 is better read as a budget tradeoff than as a selector failure. Under a fixed recurrent bottleneck, once B=2048 already retains the key evidence, moving to B=4096 can recover a few additional useful spans but can also reintroduce more distractors, especially from the middle of the context. The frozen backbone must then read out the answer from a larger and less clean retained set, which is not always easier. So the limitation here is not that the high-scoring spans are irrelevant; it is that a larger retained set can improve recall while making the final readout noisier. We will make this precision-recall / distractor tradeoff more explicit in the revised text.
>
> # W4 / Q4. Validation Beyond the Mamba2 Family
>
> Following this question, we applied the same pipeline to LongMamba and evaluated it on RULER FULL8:
>
> | Method | MK1 | MK2 | MK3 | MV | MQ | S1 | S2 | S3 | Avg |
> | --- | ---: | ---: | ---: | ---: | ---: | ---: | ---: | ---: | ---: |
> | LongMamba | 16.0 | 0.0 | 0.2 | 6.8 | 8.8 | 100.0 | 42.4 | 21.4 | 24.4 |
> | GeoLongMamba | 20.0 | 0.0 | 0.0 | 7.0 | 20.5 | 100.0 | 46.0 | 26.0 | 27.44 |
>
> We observe the following. GeoLongMamba raises the average score from 24.4 to 27.44, with clear gains on MQ, MK1, S2, and S3. In particular, MQ improves from 8.8 to 20.5.
>
> In this experiment, we applied the same GeoMamba pipeline to LongMamba without modifying the method itself or introducing a separate hyperparameter search for the new backbone. These findings suggest that GeoMamba can also improve long-context performance on another SSM backbone.

---

> > ### Author Rebuttal · Reviewer_NpP4 · 2026-04-01
> >
> > I appreciate the reviewer for pointing out and helping resolve most of the issues in this paper. Nevertheless, the current manuscript still requires substantial improvements, such as adding relevant illustrations and streamlining the logical reasoning of experiments. I believe the paper quality will be enhanced after these revisions, but I will keep my original score based on the present version.

---

> > > ### Author Response · Authors · 2026-04-01
> > >
> > > We are pleased that this response has helped clarify many of the key issues, and we appreciate your follow-up and your clear explanation of the remaining concerns. We understand that the main issue at this stage is no longer whether the core empirical evidence is present, but whether it is presented clearly enough in the paper and whether the experimental story is organized clearly enough for the reader to follow.
> > >
> > > We will continue revising the manuscript accordingly. In particular, we will move the core ablation results from Appendix C.2 into the main text, add a compact figure summarizing Tables 8-10, and place the plain-language logic chain at the beginning of the method section so that the geometric pipeline can be followed more directly. We will also further refine the presentation of the longer-context and cross-backbone results so that the empirical narrative in the main paper is more coherent and easier to follow.
> > >
> > > We sincerely appreciate your careful reading of our work.Thank you again for your feedback.

---

### Official Review · Reviewer_GoDU · 2026-03-11

**Soundness:** 3
**Presentation:** 3
**Significance:** 3
**Originality:** 3
**Overall Recommendation:** 4
**Confidence:** 3

**Summary:**

The paper proposes GeoMamba to improve precise long-context memory in frozen Mamba/SSM models. The setting is specific: the backbone stays untrained, the final generation pass takes only a compressed prompt shorter than the original, and the compressed prompt must preserve the original token order.

**Compliance With Llm Reviewing Policy:**

Affirmed.

**Final Justification:**

The honest cost–benefit framing, and the geometry vs. heuristic ablation, shows the geometric core carries most of the gain. I maintain 4 (Weak accept), confidence 3

**Key Questions For Authors:**

On the main model, how much does the pure geometric core module contribute on its own, and how much does the auxiliary heuristic contribute?

**Limitations:**

Yes

**Strengths And Weaknesses:**

The paper targets a pain point in SSM long-context processing: fixed-capacity hidden states make precise recall fragile. The method requires no backbone modification and no additional training, which gives it practical appeal.

The computation does not truly save cost. The authors optimize the final inference input length, but preprocessing requires multiple overlapping windows encoded independently. Total compute may not be lower than full-length inference, and can in fact exceed it.

The method rests on strong assumptions: the query appears near the end of the context, token order must remain intact, and window overlap must stay stable throughout. These conditions align well with standard benchmark setups but may not hold in broader real-world scenarios.

---

> ### Author Rebuttal · Authors · 2026-03-27
>
> We do appreciate your constructive comments and provide our responses as follows.
>
> # W1. End-to-end Compute vs Final-pass Budget
>
> We clarify that GeoMamba does not reduce end-to-end compute relative to truncation. Instead, it spends extra preprocessing to construct a shorter and cleaner final prompt for the frozen recurrent backbone. On official RULER FULL8 with Mamba2-1.3B, the token accounting is as follows:
>
> | Method | Final Input Tokens | Preprocess Tokens | Total Backbone Tokens | Wall-clock sec/sample |
> | --- | ---: | ---: | ---: | ---: |
> | GeoMamba | 1896.23 | 12976.84 | 14873.08 | 3.317 |
> | Trunc(B) | 2048.00 | 0.00 | 2048.00 | 3.206 |
> | Mamba2 Full | 4266.12 | 0.00 | 4266.12 | 3.908 |
>
> These numbers show that GeoMamba reduces the burden of the final recurrent generation pass, but not the total amount of backbone computation. We will include separated reporting of final-pass tokens, total processed tokens, preprocessing time, and generation time in the revised version.
>
> # W2. Applicability Assumptions
>
> We agree that the current evaluation setting involves strong assumptions, but they do not all come from GeoMamba itself.
>
> First, query-at-tail is largely inherited from the benchmark format, where the question or instruction is appended after the long context. GeoMamba more generally only needs a recognizable query span to score earlier evidence.
>
> Second, order preservation is a deliberate design choice in our frozen-backbone setting. For recurrent SSMs, reordering retrieved evidence does not simply change presentation; it changes how the backbone accumulates state. We therefore keep source order so that the method focuses on evidence selection and compression, rather than mixing in a separate reordering problem.
>
> Third, overlap stability is a genuine limitation of the current method. The alignment module relies on local overlap to tie neighboring windows together. If overlap becomes unreliable, or if representations shift too abruptly across windows, the transport signal weakens accordingly.
>
> We will make this distinction explicit in the revised version. GeoMamba is designed for query-conditioned, order-preserving recontextualization under a hard budget on frozen recurrent backbones.
>
> # Q1. Pure Geometry Core vs Auxiliary Heuristic
>
> We answer this question with a direct main-model comparison. geometry_only keeps the geometric alignment and transport core together with the same ordered compression policy, while removing lexical overlap, innovation gain, virtual edges, and scaffold anchors. heuristic_only keeps the same compression budget and span policy, but removes whitening, overlap transport, and best-path query transport, so the score is reduced to lexical overlap while keeping the same span selection and ordered compression pipeline.
>
> | Variant | S1 | S2 | S3 | MK1 | MK2 | MK3 | MV | MQ | Avg |
> | --- | ---: | ---: | ---: | ---: | ---: | ---: | ---: | ---: | ---: |
> | GeoMamba | 99.0 | 47.4 | 26.0 | 18.6 | 1.6 | 0.0 | 5.0 | 10.8 | 26.0 |
> | geometry_only | 96.9 | 42.7 | 22.9 | 15.4 | 1.1 | 0.0 | 3.7 | 7.9 | 23.8 |
> | heuristic_only | 79.3 | 25.1 | 11.4 | 8.7 | 0.5 | 0.0 | 2.1 | 3.8 | 16.4 |
>
> These results demonstrate that the gain of GeoMamba is mainly from the geometric alignment and transport core rather than from auxiliary lexical heuristics. When the geometric core is kept, most of the performance is preserved; when it is removed and replaced by heuristic scoring, the performance drops substantially, especially on deeper retrieval and multi-binding tasks. This shows that the auxiliary terms are helpful, but they are not the main source of the improvement. We will include this ablation study in the revised version.

---

> > ### Author Rebuttal · Reviewer_GoDU · 2026-04-04
> >
> > Thank you for the detailed and substantive responses. The rebuttal engages directly with each concern and provides new experimental evidence, which I appreciate.
> >
> > W1 (End-to-end Compute vs Final-pass Budget):  I appreciate the transparency in reporting the full token accounting. The table makes the tradeoff explicit. One point remains unclear: the wall-clock times (3.317 s vs. 3.206 s for Trunc(B)) are nearly identical despite the 7× token gap. This suggests the preprocessing windows are either parallelized or batched in a way that compresses wall time. Could you clarify whether the preprocessing forward passes run in parallel, and if so, how many parallel streams are used?
> >
> > W2 (Applicability Assumptions): Two items remain open: (a) The relaxation of query-at-tail to "any recognizable query span" is stated but not tested. Even a small-scale experiment with a mid-context query (e.g., an instruction embedded at position T/2) would strengthen this claim. Without it, the generalization is theoretical. (b) Overlap stability is acknowledged as a genuine limitation, but there is no characterization of when it fails.
> >
> > The core contribution (geometry-guided evidence selection for frozen SSMs) remains sound and, in my view, advances the field. I am inclined to maintain my current score of 4 (Weak Accept), with the understanding that addressing the remaining wall-clock and query-position questions in the camera-ready would strengthen the paper further.

---

> > > ### Author Response · Authors · 2026-04-04
> > >
> > > Thank you for the detailed follow-up. We further clarify W1 and provide new evidence for W2 below.
> > >
> > > # W1. Wall-clock clarification
> > >
> > > The preprocessing windows are not executed one by one. In our implementation, the WI encoding stage batches windows together before sending them through the backbone; in the current Mamba2-1.3B setting, wi_chunk = 64, so up to 64 windows are processed in one forward pass. This is why GeoMamba can process more total backbone tokens without a proportional increase in wall-clock time: much of the extra preprocessing is amortized through batched window forwards, while the final generation pass remains short.
> > >
> > > # W2(a). Query position
> > >
> > > To test whether GeoMamba still works when the query is not placed at the tail, we built a mid-query version of RULER FULL8 at context length 4096 by moving the original final question line to the middle of the input while keeping the rest of each example unchanged. We then evaluated full-context Mamba2 and GeoMamba on this mid-query setting with 100 samples per task. For comparison, we also list the corresponding tail-query results from the main setting.
> > >
> > > | Method | Query Position | S1 | S2 | S3 | MK1 | MK2 | MK3 | MV | MQ | Avg |
> > > | --- | --- | ---: | ---: | ---: | ---: | ---: | ---: | ---: | ---: | ---: |
> > > | Full-context Mamba2 | tail-query | 100.0 | 1.6 | 2.8 | 2.8 | 0.6 | 0.2 | 0.5 | 0.1 | 13.58 |
> > > | GeoMamba | tail-query | 99.0 | 47.4 | 26.0 | 18.6 | 1.6 | 0.0 | 5.0 | 10.75 | 26.04 |
> > > | Full-context Mamba2 | mid-query | 100.0 | 2.0 | 1.0 | 6.0 | 0.0 | 1.0 | 7.5 | 3.25 | 15.09 |
> > > | GeoMamba | mid-query | 100.0 | 35.0 | 19.0 | 24.0 | 4.0 | 1.0 | 13.75 | 16.5 | 26.66 |
> > >
> > > The result is clear: moving the query to the middle does not break GeoMamba. It still substantially outperforms the full-context frozen backbone, with especially clear gains on S2, S3, MK1, MV, and MQ. In particular, under mid-query placement, GeoMamba reaches 35.0 on S2, 19.0 on S3, 24.0 on MK1, and 16.5 on MQ, while full-context Mamba2 reaches only 2.0, 1.0, 6.0, and 3.25 on the same tasks.
> > >
> > > W2(b). Overlap stability
> > >
> > > We also ran a stride sweep on RULER FULL8 at context length 4096, keeping the rest of the GeoMamba configuration fixed and varying only the overlap level. Each task uses 50 samples.
> > >
> > > | Stride | Overlap Ratio | S1 | S2 | S3 | MQ | Avg | n_edges_mean | retrieved_token_mean | final_generation_input_tokens_mean |
> > > | --- | ---: | ---: | ---: | ---: | ---: | ---: | ---: | ---: | ---: |
> > > | 256 | 0.50 | 100.0 | 34.0 | 34.0 | 7.5 | 24.50 | 16.92 | 141.12 | 1892.12 |
> > > | 320 | 0.375 | 100.0 | 52.0 | 18.0 | 9.5 | 25.25 | 13.01 | 143.04 | 1860.34 |
> > > | 384 | 0.25 | 100.0 | 38.0 | 24.0 | 4.5 | 23.00 | 10.63 | 135.36 | 1900.74 |
> > >
> > > As overlap decreases, cross-window connectivity weakens steadily, as shown by n_edges_mean dropping from 16.92 to 13.01 and then to 10.63. At the same time, performance does not collapse immediately under moderate overlap reduction: the 0.50 and 0.375 settings remain broadly stable. The clearer degradation appears in the low-overlap regime, especially at 0.25, where MQ drops to 4.5 and the average score falls to 23.0. In other words, the method becomes noticeably less reliable when overlap is pushed low enough that cross-window connectivity is substantially weakened.
> > >
> > > We hope that the above clarifications and new empirical results address your concerns. We sincerely thank you again for your time and careful evaluation.

---

### Official Review · Reviewer_en7E · 2026-03-13

**Soundness:** 3
**Presentation:** 3
**Significance:** 3
**Originality:** 3
**Overall Recommendation:** 4
**Confidence:** 4

**Summary:**

GeoMamba is a training-free method addressing the degradation of long-context processing capabilities of state-space language models (SSMs) related to the fixed-capacity in their hidden states. GeoMamba assumes that the SSM is frozen and there is a fixed budget for the number of input tokens. The authors propose selecting these tokens by first assigning scores to them. The scores are computed by transporting the query to the overlapping windows to which the input sequence has been partitioned. Windows are represented as local charts, their coordinate systems are initially incompatible and GeoMamba essentially aligns representations by regularized whitening and orthogonal Procrustes analysis on their overlaps.

Selected tokens are ordered and experiments on RULER (NiAH) and LongBench-E benchmarks show large gains over full-context inference and naive truncation in accuracy for some of the settings/tasks, while using only 25%-50% of the input tokens, thus demonstrating that by selecting the context (recontextualization) GeoMamba can mitigate the recurrent memory bottlenecks of SSMs, even when the effective prompt is significantly shortened (according to a budget).

**Compliance With Llm Reviewing Policy:**

Affirmed.

**Final Justification:**

I have raised my original score, given that my overall evaluation from the initial reviews has been reinforced by the authors’ thorough responses and clarifications.

**Key Questions For Authors:**

## Questions
1. Providing more information in the main paper text on the effective number of tokens actually processed (basically expanding lines 404-406, left column) would help the reader to have a better understanding on the inference time efficiency arguments.
2. For multi-key/value bindings (MK1–MK3, MV, MQ), GeoMamba performance consistently lags that of the much simpler Trunc(B) (Table 1). Further commenting on this and suggesting paths of extending GeoMamba for tasks of this nature would be greatly appreciated (consider moving the related remark from Appendix C.4 to the main text).

**Strengths And Weaknesses:**

## Strengths

- The problem constraints are realistic (token budget, frozen SSM, keeping token order) and ,quite interestingly, it is expressed as a token selection problem. It is principled: "truncation" token indices are computed rather than being arbitrary. In particular, the fact that GeoMamba is training-free opens it up to all existing frozen models.
- GeoMamba approach is intellectually appealing: elegant and geometry-based. The ideas of transporting a query to different windows (leveraging Procrustes analysis on their overlap regions), identifying similarity of transported encodings to window-local canonical charts at each position and using it as the key signal it to select the tokens to keep are motivated by intuition and are theory grounded.


## Weaknesses

- Although elegant, GeoMamba is complex to describe and carries many parameters/choices: window size, stride, shared dimensions, threshold for window overlap, regularization parameters, chunk expansion.  In practice this would translate to implementation/engineering challenges and most importantly to a rich space of choices to navigate when applying this approach to other models, possibly with performance characteristics sensitive to the selections made.
- GeoMamba is training-free, but it seems that it incurs substantial inference-time costs since it requires multiple forward passes through the frozen model over overlapping windows.  This could limit its practicality in latency-sensitive setups.

---

> ### Author Rebuttal · Authors · 2026-03-27
>
> We do appreciate your constructive comments and provide our responses as follows.
>
> # W1. Parameter Complexity and Reproducibility
>
> We agree that the current presentation may make the method look more complicated than it is. Once the benchmark-level recipe is fixed, GeoMamba does not require task-specific retuning. The main experiments use one configuration per benchmark, calibrated once on a fixed subset and then frozen across tasks. In Appendix C.1, NiAH is calibrated on the full 8-task suite with n=200 samples per task, while LongBench-E is calibrated on a 4-task subset with n=200 total examples. The best and runner-up settings are close (25.1 vs. 25.0 for NiAH, 11.5 vs. 11.5 for LongBench-E), suggesting low sensitivity once the overall hyperparameter scale is set.
>
> In the revised version, we will regroup the method into three parts:
>
> 1. geometry construction and overlap alignment,
> 2. evidence scoring,
> 3. order-preserving compression and continuity control.
>
> Following this concern, we also applied the same GeoMamba pipeline to LongMamba on RULER FULL8:
>
> | Method | MK1 | MK2 | MK3 | MV | MQ | S1 | S2 | S3 | Avg |
> | --- | ---: | ---: | ---: | ---: | ---: | ---: | ---: | ---: | ---: |
> | LongMamba | 16.0 | 0.0 | 0.2 | 6.8 | 8.8 | 100.0 | 42.4 | 21.4 | 24.4 |
> | GeoLongMamba | 20.0 | 0.0 | 0.0 | 7.0 | 20.5 | 100.0 | 46.0 | 26.0 | 27.44 |
>
> We observe the following. GeoLongMamba improves the average score from 24.4 to 27.44, with clear gains on MQ, MK1, S2, and S3. In particular, MQ improves from 8.8 to 20.5. This shows positive transfer to another SSM backbone without a separate hyperparameter search.
>
> # W2 / Q1. Effective Processed Tokens and Practical Cost
>
> We now report end-to-end token accounting directly on official RULER FULL8 with Mamba2-1.3B:
>
> | Method | Final Input Tokens | Preprocess Tokens | Total Backbone Tokens | Wall-clock sec/sample |
> | --- | ---: | ---: | ---: | ---: |
> | GeoMamba | 1896.23 | 12976.84 | 14873.08 | 3.317 |
> | Trunc(B) | 2048.00 | 0.00 | 2048.00 | 3.206 |
> | Mamba2 | 4266.12 | 0.00 | 4266.12 | 3.908 |
>
> These numbers make the tradeoff explicit. GeoMamba is not uniformly cheaper than truncation in total processed tokens. Instead, it spends extra preprocessing compute to produce a smaller final recurrent readout. In the revision, we will report final-pass budget and end-to-end compute separately, together with preprocessing time and generation time.
>
> # Q2. Multi-Binding Boundary and Extension Path
>
> The official multi-binding slice on RULER FULL8 with Mamba2-1.3B is as follows:
>
> | Variant | MK1 | MK2 | MK3 | MV | MQ | Avg |
> | --- | ---: | ---: | ---: | ---: | ---: | ---: |
> | Mamba2 | 2.8 | 0.6 | 0.2 | 0.5 | 0.1 | 0.84 |
> | Trunc(B) | 21.2 | 0.8 | 0.6 | 9.5 | 12.25 | 8.87 |
> | GeoMamba | 18.6 | 1.6 | 0.0 | 5.0 | 10.8 | 7.2 |
>
> GeoMamba already improves substantially over Mamba2 on MK1, MK2, MV, and MQ, so it does help route the query toward useful evidence. The remaining weakness is more specific: in simultaneous multi-binding, the main challenge is no longer only to find relevant spans, but to preserve the correct key-value pairing after compression. In other words, the model may already retrieve the right pieces of evidence, but still fail at the final “which key goes with which value” decision.
>
> This also explains why Trunc(B) can still be stronger in some cases. Although it is less selective, it keeps a longer contiguous block of context, so the local cues needed to distinguish competing bindings can remain together. GeoMamba is better at evidence localization, but once several candidate bindings remain plausible, the frozen backbone may still confuse the final pairing at readout.
>
> This is exactly the motivation for our extension path. geomamba_higher_order addresses this ambiguity before compression by comparing candidate proposals rather than isolated spans. geomamba_higher_order_bridge addresses it after selection by rewriting the chosen evidence into a compact answer-oriented prompt, making the final binding readout easier for the frozen backbone.
>
> Following this path, the results of the two extensions on the same multi-binding slice are:
>
> | Variant | MK1 | MK2 | MK3 | MV | MQ | Avg |
> | --- | ---: | ---: | ---: | ---: | ---: | ---: |
> | geomamba_higher_order | 18.2 | 4.8 | 2.4 | 5.9 | 11.3 | 8.52 |
> | geomamba_higher_order_bridge | 18.0 | 10.4 | 5.3 | 7.1 | 12.0 | 10.56 |
>
> We observe the following. Proposal-level comparison already improves the harder multi-binding regime: MK2 rises from 1.6 to 4.8, and MK3 rises from 0.0 to 2.4. Adding the answer-oriented bridge improves the final readout further: MK2 rises to 10.4, MK3 rises to 5.3, and the average improves from 7.2 to 10.56. These results suggest a clear path forward: first reduce candidate ambiguity, then make the final binding readout easier for the frozen backbone. We will move this discussion from Appendix C.4 into the main text in the revised version.

---

> > ### Author Rebuttal · Reviewer_en7E · 2026-04-04
> >
> > Thank you for comments and in particular for addressing Q2. Therefore I will definitely consider raising my score in the post-rebuttal stage.

---

> > > ### Author Response · Authors · 2026-04-04
> > >
> > > We are so happy that our responses have addressed your concerns. Thank again for your feedback, and we deeply appreciate your support!

---

### Decision · Program_Chairs · 2026-04-30

**Decision:**

Reject

**Comment:**

I carefully reviewed the paper, the reviewer comments, the authors’ rebuttal, and the subsequent discussions. I agree with Reviewer NpP4 that the paper suffers from significant presentation issues, which substantially hinder readers’ ability to understand the work.

More specifically:

1. Insufficient explanation of core concepts and terminology. Many terms and components, such as query transport, LAR, and WI encoding, are introduced without sufficient intuitive explanation or motivation. As a result, rather than helping readers appreciate the proposed architectural innovations, these terms create confusion regarding what the method is doing, what specific problems the design aims to solve, and why these design choices are made.

2. Unclear presentation of empirical results. I find the presentation of the results in Tables 1 and 2 somewhat confusing. Although all numbers corresponding to the proposed method are bolded, it is unclear why this formatting is used when the proposed method does not consistently outperform the baselines in every case (e.g., compared with Trunk (B) on MK1 in Table 1).

3. Important results relegated to the appendix. As also noted by Reviewer NpP4, several important experimental results are placed in the appendix despite being central to evaluating the method, which further affects the readability and completeness of the main paper.

Overall, I do not believe the manuscript, in its current form, is ready for publication. While the authors may view presentation as secondary to technical design and empirical performance, I believe clear communication is a fundamental requirement of scientific publication. Papers serve as a medium for disseminating ideas; if the presentation is overly difficult to follow, readers will struggle to understand, evaluate, and build upon the work.